# The flavonoid corylin exhibits lifespan extension properties in mouse

Tong-Hong Wang [1,2,15], Wei-Che Tseng [3,15], Yann-Lii Leu[1,3,15], Chi-Yuan Chen[1,2], Wen-Chih Lee [4], Ying-Chih Chi[5], Shu-Fang Cheng[3], Chun-Yu Lai[3], Chen-Hsin Kuo[3], Shu-Ling Yang[6], Sien-Hung Yang[6,7], Jiann-Jong Shen[6], Chun-Hao Feng[3], Chih-Ching Wu [8,9,10,11], Tsong-Long Hwang [2,3,12], Chia-Jen Wang[13], Shu-Huei Wang [14,16 ✉] & Chin-Chuan Chen [1,3,16 ✉]

In the long history of traditional Chinese medicine, single herbs and complex formulas have been suggested to increase lifespan. However, the identification of single molecules responsible for lifespan extension has been challenging. Here, we collected a list of traditional Chinese medicines with potential longevity properties from pharmacopeias. By utilizing the mother enrichment program, we systematically screened these traditional Chinese medicines and identified a single herb, *Psoralea corylifolia*, that increases lifespan in Saccharomyces cerevisiae. Next, twenty-two pure compounds were isolated from *Psoralea corylifolia*. One of the compounds, corylin, was found to extend the replicative lifespan in yeast by targeting the Gtr1 protein. In human umbilical vein endothelial cells, RNA sequencing data showed that corylin ameliorates cellular senescence. We also examined an in vivo mammalian model, and found that corylin extends lifespan in mice fed a high-fat diet. Taken together, these findings suggest that corylin may promote longevity.

[1] Tissue Bank, Chang Gung Memorial Hospital, Taoyuan, Taiwan. [2] Graduate Institute of Health Industry Technology, Research Center for Food and Cosmetic Safety, Research Center for Chinese Herbal Medicine, College of Human Ecology, Chang Gung University of Science and Technology, Taoyuan, Taiwan. [3] Graduate Institute of Natural Products, Chang Gung University, Taoyuan, Taiwan. [4] Office of Research and Development, Tzu Chi University, Hualien, Taiwan. [5] Cryo-EM Center, Vagelos College of Physicians and Surgeons, Columbia University Irving Medical Center, New York, USA. [6] School of Traditional Chinese Medicine, Chang Gung University, Taoyuan, Taiwan. [7] Division of Chinese Internal Medicine, Center for Traditional Chinese Medicine, Chang Gung Memorial Hospital, Taoyuan, Taiwan. [8] Department of Medical Biotechnology and Laboratory Science, Chang Gung University, Taoyuan, Taiwan. [9] Department of Otolaryngology-Head & Neck Surgery, Chang Gung Memorial Hospital, Taoyuan, Taiwan. [10] Molecular Medicine Research Center, Chang Gung University, Taoyuan, Taiwan. [11] Research Center for Emerging Viral Infections, College of Medicine, Chang Gung University, Taoyuan, Taiwan. [12] Department of Anesthesiology, Chang Gung Memorial Hospital, Taoyuan, Taiwan. [13] Cell Therapy Core Laboratory, Chang Gung Memorial Hospital, Taoyuan, Taiwan. [14] Department of Anatomy and Cell Biology, College of Medicine, National Taiwan University, Taipei, Taiwan. [15]These authors contributed equally: Tong-Hong Wang, Wei-Che Tseng, Yann-Lii Leu. [16]These authors jointly supervised this work: Shu-Huei Wang, Chin-Chuan Chen. ✉email: shwang@ntu.edu.tw; chinchuan@mail.cgu.edu.tw

Aging is an irreversible functional decline that occurs in all organisms. Emerging evidence showed that senescent cell accumulation in tissues and organs deteriorates biological function and possibly contributes to aging-associated pathology[1]. Indeed, the dysregulated molecular mechanisms of aging are highly associated to aging diseases. It has been reported that pharmaceutical compounds that extend lifespan can also be used to diabetes and cardiovascular diseases[2]. Therefore, discovering antiaging drugs could provide both benefits to healthy aging and therapeutics for aging-related diseases. In the past two decades, the TOR complex and SIRT1 activities were shown to play regulatory roles in lifespan. In particular, the EGO complex regulates mTOR1 signaling in response to amino acids abundance[3]. Growth factors regulate the mTOR1 pathway through PI3K and AKT[4]. Accordingly, caloric restriction and dietary restriction are proposed to inactivate mTOR1 signaling, resulting in lifespan extension[2]. Moreover, SIRT1 has been suggested to enhance mitochondria and alter metabolic rates to increase lifespan through its HDAC activity[5]. Despite such extensive studies in the field of aging, the number of compounds identified by intervention testing programs that extend the lifespan in mice is very small[6].

Scientists have been eager to identify pharmaceuticals that can extend lifespan and prevent aging diseases throughout history. However, only a few compounds with such activities have been described. There are only limited means to screen lifespan-extending drugs[7,8]. Therefore, developing feasible and efficient methods for such screenings remains an important obstacle. Of the available screening methods for lifespan-extending drugs, the SIRT1 in vitro assay allows relatively simple and rapid data collection[9]. However, there are limitations of this method[10]. Budding yeast represents one of the simplest and widely adopted model organisms for studying aging. Relative to other systems, the lifespan of yeast can be easily quantified. Micromanipulation is the most widely used method for assessing the replicative lifespan (RLS) of yeast; however, this is an impractical drug screen method due to its time consuming process and technical difficulty on analyzing lots of sample[11]. The mother enrichment program (MEP) has been developed to monitor RLS, and it provides an alternative and simple screening method for examining replicative aging from liquid culture. This method utilizes genetic manipulation by expressing estradiol-dependent Cre recombinase (Cre-EBD78) with a daughter-specific promoter derived from SCW11 ($P_{SCW11}$). Two essential genes, UBC9 and CDC20, were then conditionally disrupted by the Cre-lox recombination system in the daughter cells in the presence of estradiol. The daughter cells were arrested by estradiol culture; thus, plating the culture liquid onto an estradiol-free plate allows only mother cells to replicate and form colonies. This system created an effective way to monitor mother cells without daughter cell disruption[12]. Compared with micromanipulation, the MEP assay provides a relatively quick assessment of the yeast RLS, and a large sample size can be analyzed. Here, we adopted this method and used its advantages to identify lifespan extending compounds.

Several traditional Chinese medicines (TCMs) may contribute to lifespan extension benefits according to TCM pharmacopeias, such as the Compendium of Materia Medica, Qianjinyaofang, Shennong Materia Medica, and Huangdi Neijing. However, the key compounds from these TCMs have not been purified, and their lifespan-extending activities have not been validated[13–16]. Here, we utilized the MEP system to identify crude extracts from one of these TCMs, Psoralea corylifolia (P. corylifolia), which has great potential to extend lifespan. In addition, we purified the n-hexane-soluble fraction of P. corylifolia and identified twenty-two compounds by NMR, FT-IR, UV, and mass spectral analysis. By taking advantage of the MEP assay, we validated the activities of these compounds and identified two active compounds from P. corylifolia that may increase lifespan: corylin and neobavaisoflavone. In addition, we demonstrated that corylin docking to Gtr1 and therefore suppresses the Tor1 activity, which contributes to lifespan extension. In mammals, cellular senescence is proposed to cause physical dysfunction and exacerbate aging process[17,18]. It is suggested that loss of DNA repair capacity, chromosome instability, and telomere erosion cause cell senescence. Additionally, senescent cells produce a senescence-associated secretory phenotype (SASP) to broaden the impact by triggering the senescence process in the ambient tissue[1]. Finally, senescent cells deteriorate organ function and trigger aging-related disease, even raising the risk of death[18].

In this work, we demonstrated that corylin alleviates senescence process in HUVECs through suppressing the mTOR pathway to increase lifespan. A previous study revealed that a high-fat diet (HFD) will induce metabolic stress and accelerate senescent cell accumulation to further increase the risk of death[18]. We further demonstrate that corylin promotes longevity in aged mice under metabolic stress, which likely contributed by improving physical functions, reducing metabolic stress, and maintaining tissue functional markers.

## Results

**Validation of TCMs for lifespan extension**. Aging research has been an active field in recent decades, however, due to the limitation regarding screening methods and the insufficient number of candidates. To date, only a few compounds have been rigorously identified to extend lifespan[19]. In the long history of Chinese medicine, some TCMs have been reported to have lifespan-extending benefits. Thus, TCMs are great candidates for screening and verifying their life-extending properties. To choose TCM candidates for validation, we searched the National Health Insurance Research Database (NHIRD) as well as several pharmacopeias, such as the Compendium of Materia Medica, Qianjinyaofang, Shennong Materia Medica, and Huangdi Neijing, for TCMs that may increase lifespan and are used to treat age-associated diseases, for example, osteoporosis, sarcopenia, and dementia. After cross-comparison of these TCMs and consultation with clinical TCM doctors, 33 single TCM herbs and six TCM herbal formulas were chosen as candidates (Table 1).

***Psoralea corylifolia* increases viability in mother enrichment program (MEP)**. The MEP was developed to monitor the RLS of mother cells from liquid cultures[12]. We first investigated whether the TCMs listed in Table 1 extend the RLS of yeast using the MEP assay. Based on crude water and ethanol extractions (Supplementary Fig. 1), we generated 78 candidates from that list. Each sample listed in Table 1 was tested using the MEP assay (Supplementary Figs. 2 and 3). We found that the ethanol extract of P. corylifolia significantly increased viability based on MEP, suggesting that the ethanol extract of P. corylifolia may extend the RLS of yeast (Fig. 1a).

***Psoralea corylifolia* increases the RLS of yeast**. The MEP assay offers an easy and efficient strategy for assessing yeast RLS. However, once the division rates change with the age of yeast, the TCM could distort the MEP viability curve[12]. Thus, we sought to identify whether P. corylifolia increases yeast RLS in a micromanipulation system. Consistent with the MEP screening, the ethanol extract of P. corylifolia at 10 μg/ml significantly extended the RLS by micromanipulation (Fig. 1b). P. corylifolia is mainly distributed throughout India and South Asia. The different distributions under different cultivation conditions could have distinct chemical constituents. To characterize P. corylifolia, the

fingerprint of the chemical constituents in the ethanol fractions of *P. corylifolia* was determined by HPLC. Under the optimal chromatographic conditions, the major components of *P. corylifolia* were identified, and baseline separation was obtained. Fig. 1c shows a representative chromatogram of the ethanol extract of *P. corylifolia*. The fingerprint derived from the HPLC assay is a reproducible and reliable method and could be used in the quality control of the preparation of active fractions of *P. corylifolia*.

**Table 1 Lists of TCM single herbs and herbal formulas in the TCM pharmacopeia that are suggested to treat age-related diseases and/or extend the lifespan.**

**TCM single herbs**

| Chinese name in TCM Pharmacopeia | Scientific name | Genus and trivial name |
|---|---|---|
| 女貞子 | *Ligustrum lucidum* | *Fructus Ligustri Lucidi* |
| 銀杏葉 | *Ginkgo biloba* | *Folium Ginkgo* |
| 薑黃 | *Curcuma longa* | *Rhizoma Curcumae Longae* |
| 半夏 | *Pinellia ternata* | *Pinellia Tuber* |
| 覆盆子 | *Rubus idaeus* | *Fructus Rubi* |
| 天麻 | *Gastrodia elata* | *Rhizoma Gastrodiae* |
| 桂枝 | *Cinnamomum cassia* | *Ramulus Cinnamomi* |
| 石斛 | *Dendrobium nobile* | *Caulis Dendrobii* |
| 貝母 | *Fritillaria thunbergii* | *Bulbus Fritillariae* |
| 甘草 | *Glycyrrhiza uralensis* | *Radix Glycyrrhizae* |
| 栝蔞根 | *Trichosanthes kirilowii* | *Radix Trichosanthis* |
| 茯苓 | *Poria cocos* | *Poria* |
| 枳實 | *Citrus aurantium* | *Fructus Aurantii Immaturus* |
| 紫蘇葉 | *Perilla frutescens* | *Folium Perillae* |
| 薏苡仁 | *Coix lacryma-jobi* | *Semen Coicis* |
| 紫蘇梗 | *Perilla frutescens* | *Caulis Perillae* |
| 淫羊藿 | *Epimedium brevicornum* | *Herba Epimedii* |
| 何首烏 | *Polygonum multiflorum* | *Radix Polygoni Multiflori* |
| 黃耆 | *Astragalus membranaceus* | *Radix Astragali* |
| 杜仲 | *Eucommia ulmoides* | *Cortex Eucommiae* |
| 豬苓 | *Polyporus umbellatus* | *Polyporus* |
| 刺五加 | *Eleutherococcus senticosus* | *Radix Acanthopanacis Senticosl* |
| 絞股藍 | *Gynostemma pentaphyllum* | *Herba Gynostemmatis Pentaphylli* |
| 補骨脂 | *Psoralea corylifolia* | *Fructus Psoraleae* |
| 菟絲子 | *Cuscuta chinensis* | *Semen Cuscutae* |
| 蛇床子 | *Cnidium monnieri* | *Fructus Cnidii* |
| 粳米 | *Oryza sativa* | *Fructus Germinantus Oryzae* |
| 葛根 | *Pueraria lobata* | *Radix Puerariae* |
| 龍膽草 | *Gentiana scabra* | *Radix Gentianae* |
| 大黃 | *Rheum palmatum* | *Radix et Rhizoma Rhei* |
| 香附子 | *Cyperus rotundus* | *Rhizoma Cyperi* |
| 酸棗仁 | *Ziziphus jujuba* | *Semen Ziziphi Spinosae* |
| 白果 | *Ginkgo biloba* | *Semen Ginkgo* |

**CM herbal formula**

| Chinese name in TCM Pharmacopeia | Scientific name |
|---|---|
| 十全大補湯 | Shi Quan Da Bu Tang |
| 還少丹 | Huan Shao Dan |
| 八味地黃丸 | Ba Wei Di Huang Wan |
| 六味地黃丸 | Liu Wei Di Huang Wan |
| 左歸丸 | Zuo Gui Wan |
| 血府逐瘀湯 | Xue Fu Zhu Yu Tang |

**Psoralea corylifolia promotes yeast RLS in a Tor1-dependent manner**. There are a significant number of compounds in the crude ethanol extract of *P. corylifolia*. To narrow down the candidates, we first partitioned the ethanol crude extract into *n*-hexane and water to separate the low- and high-polarity compounds. Using the micromanipulation assay, we identified that the *n*-hexane fraction of the ethanol extract of *P. corylifolia* increased the yeast lifespan by 20% at 10 μg/ml (Fig. 1d). Next, we investigated the mechanism of the lifespan-extending activity of the *n*-hexane fraction of the ethanol extract of *P. corylifolia*. In *Saccharomyces cerevisiae*, SIR2 and TOR1 are key players regulating the lifespan of yeast. The Sir2 protein has been well characterized as a lifespan-extension factor for replicative lifespan[20]. The *sir2Δfob1Δ* double deletion strain was used to study the genetic pathway of sir2[21]. Interestingly, we found that the *n*-hexane extract of *P. corylifolia* promotes the mean lifespan of the *sir2Δfob1Δ* strain, indicating that the *n*-hexane fraction of *P. corylifolia* increases RLS via a Sir2-independent pathway (Fig. 1e). We next examined whether the *n*-hexane extract of *P. corylifolia* elevated yeast RLS through a tor1-dependent pathway. As the data show, treatment with *P. corylifolia* did not extend the RLS of the *tor1Δ* deletion strain (Fig. 1f). These data suggested that Tor1 plays a major role in the ability of the *P. corylifolia n*-hexane-soluble fraction to extend the RLS of *Saccharomyces cerevisiae*.

**Corylin and neobavaisoflavone are the active compounds in Psoralea corylifolia that extend the RLS of yeast**. Although we were able to verify that the *n*-hexane fraction of the ethanol extract of *P. corylifolia* contributes to RLS extension in yeast, the *n*-hexane fraction of *P. corylifolia* contains numerous constituents. To identify the active compound (s) from the *n*-hexane extract of *P. corylifolia* that increase yeast RLS, we first separated compounds in the *n*-hexane extract by column chromatography and isolated 22 pure compounds. Each of these compounds was identified by spectroscopic analyses, including 2D-NMR, FT-IR, and UV, as well as a literature survey (Fig. 2). Interestingly, compounds 15, 16, 17, and 18 were isolated from *P. corylifolia* for the first time. To identify the active compound that increases lifespan in *P. corylifolia*, 22 components were tested using the MEP assay (Fig. 3a–i). Due to the limited amounts of compounds, a concentration of 15 μM was chosen for the MEP assay. The flavonoid subgroup chalcones (18–22) showed cytotoxicity at 15 μM (Supplementary Fig. 4); therefore, the concentration was reduced to 1.5 μM. We found that 2 of the 22 compounds, corylin (12) and neobavaisoflavone (14), significantly increased viability in the MEP system at 15 μM (Fig. 3e, f). These results showed that corylin (12) and neobavaisoflavone (14) may be the major components responsible for the lifespan-extending ability of *P. corylifolia*. Of these two compounds, corylin more effectively increased the viability of yeast in the MEP assay compared with neobavaisoflavone (Fig. 3f). In addition, neobavaisoflavone (14) shares the same basic skeleton as corylin with few differences. Moreover, corylin was ten times more abundant than neobavaisoflavone in the *n*-hexane fraction, suggesting that corylin is the major active constituent of *P. corylifolia*.

**Corylin increases yeast RLS through the Tor pathway**. To further confirm the data obtained from the MEP assay, we next examined whether corylin promotes RLS in yeast by using a micromanipulation assay. The results showed that corylin significantly increased the RLS of yeast, indicating that corylin is one of the active compounds in *P. corylifolia* that increased yeast RLS (Fig. 4a). To identify the mechanism of corylin related to lifespan extension, we next examined whether corylin exerted its effect via

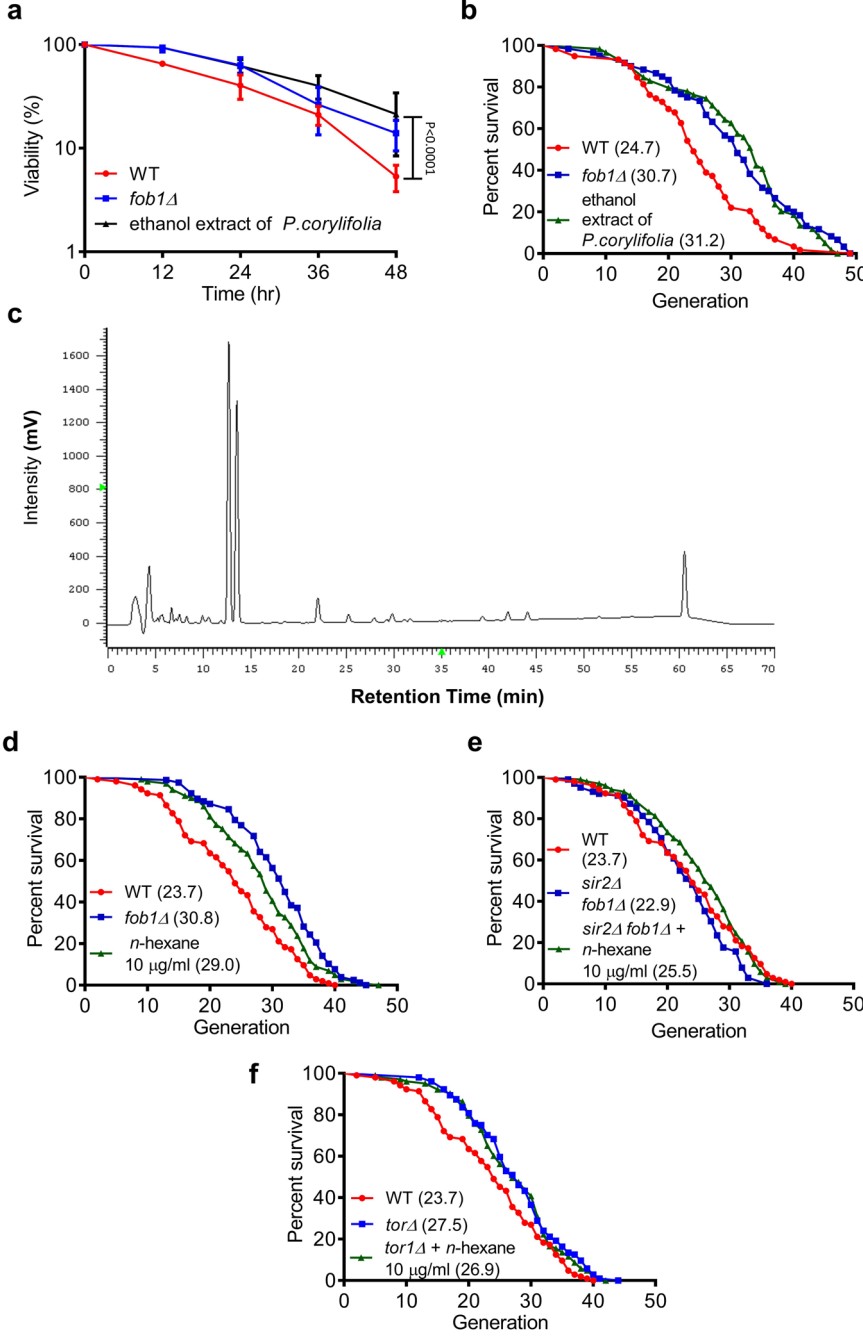

**Fig. 1 Ethanol extract of *Psoralea corylifolia* extends the replicative lifespan of *Saccharomyces cerevisiae*. a** Viability curve of the haploid MEP strain ZHY1 in liquid with a 10 μg/ml ethanol extract of *P.corylifolia* or DMSO, the mutation of *fob1Δ* is known to increase RLS ($n = 3$). **b** The RLS was determined by micromanipulating individual yeast cells on YEPD with or without the ethanol extract of *P. corylifolia*. The mean for WT = 24.7 generations ($n = 59$); *fob1Δ* = 30.7 generations ($n = 60$); and the *P. corylifolia* ethanol extract at 10 μg/ml = 31.2 generations ($n = 59$). **c** Column: MIGHTYSIL RP-18, 5 μm. Eluents: solvent A: 0.1% formic acid—acetonitrile and solvent B: 0.1% formic acid–water. Elution profile: 0–20 min, 60–50% B (40–50% A); 20–35 min, 50–40% B (50–60% A); 35–45 min, 40–30% B (60–70% A); 45–55 min, 30–20% B (70–80% A); 55–60 min, 60–50% B (40–50% A); 70 min, stop. Detection: UV at 245 nm. The RLS was determined by micromanipulating individual yeast cells on YEPD with or without the *n*-hexane extract of *P. corylifolia*. **d** The mean for WT = 23.7 generations ($n = 104$); *fob1Δ* = 30.8 generations ($n = 78$); and the *n*-hexane extract of *P. corylifolia* at 10 μg/ml = 29.0 generations ($n = 96$). **e** The mean for WT = 23.7 generations ($n = 104$); *sir2Δ fob1Δ* double deletion strain (*sir2Δ fob1Δ*) = 22.9 generations ($n = 102$); and sir2Δfob1Δ double deletion strain with the *n*-hexane extract of *P. corylifolia* at 10 μg/ml (*sir2Δ fob1Δ*+ *n*-hexane) = 25.5 generations ($n = 102$). **f** The mean for WT = 23.7 generations ($n = 104$); *tor1Δ* = 27.5 generations ($n = 104$); and *tor1Δ* with the *n*-hexane extract of *P. corylifolia* at 10 μg/ml (*tor1Δ*+ *n*-hexane) = 26.9 generations ($n = 103$). Data presented as mean ± SD from at least three biologically independent experiments. **a** *p* values were determined by two-way ANOVA (multiple comparisons). **b**, **d**–**f** *p* values were determined by the Gehan–Breslow–Wilcoxon test (see also Supplementary Fig. 14). Source data are provided as a Source data file.

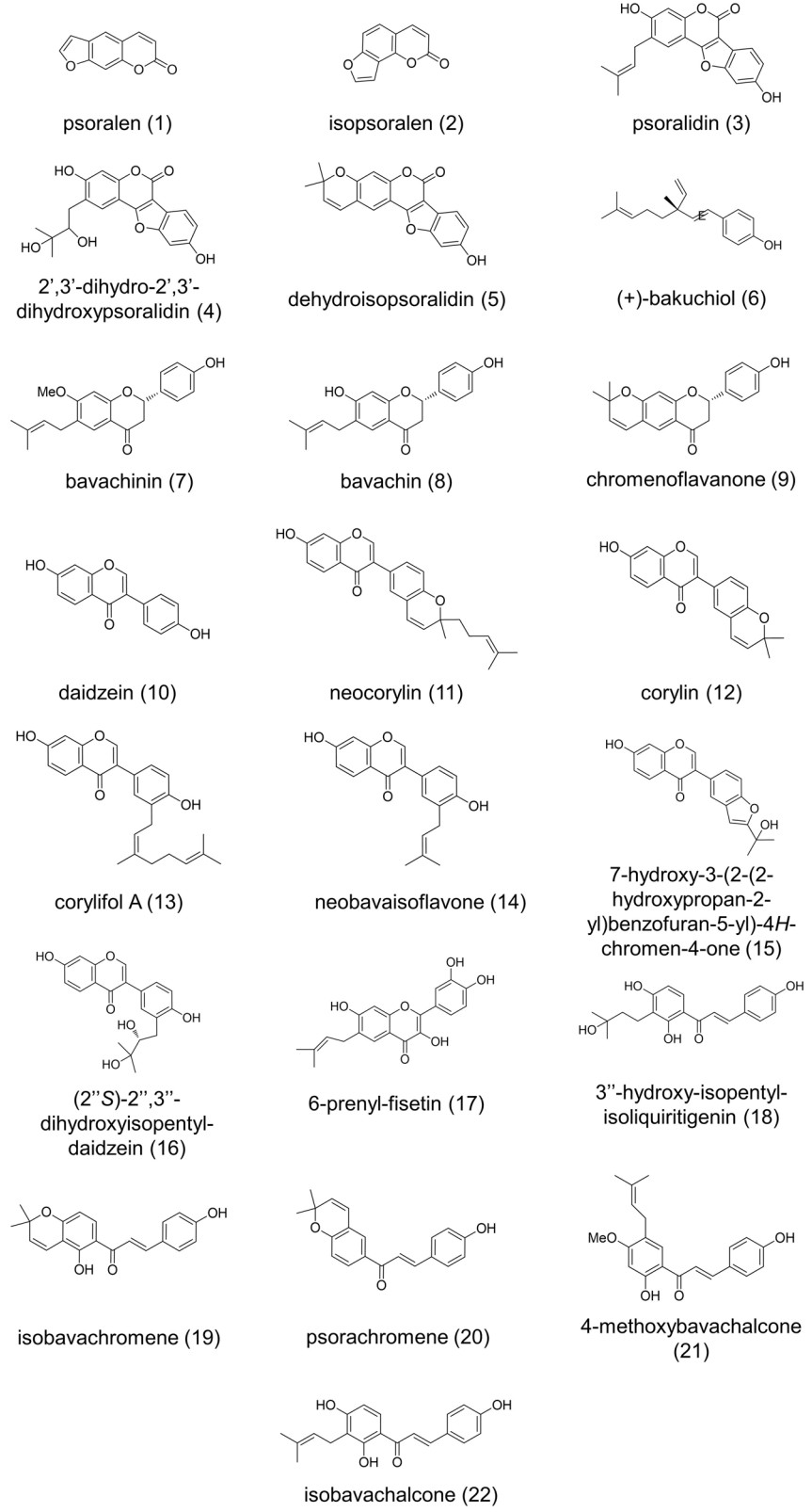

**Fig. 2 Pure compounds isolated from the *n*-hexane extract of *Psoralea corylifolia*.** The compounds were classified according to their structure: coumarins (1–5), benzenoids (6) and flavonoids (7–22). The flavonoid group was further divided into four subgroups according to their structure: flavanones (7–9), isoflavones (10–16), flavonol (17) and chalcones (18–22). Source data are provided as a Source data file.

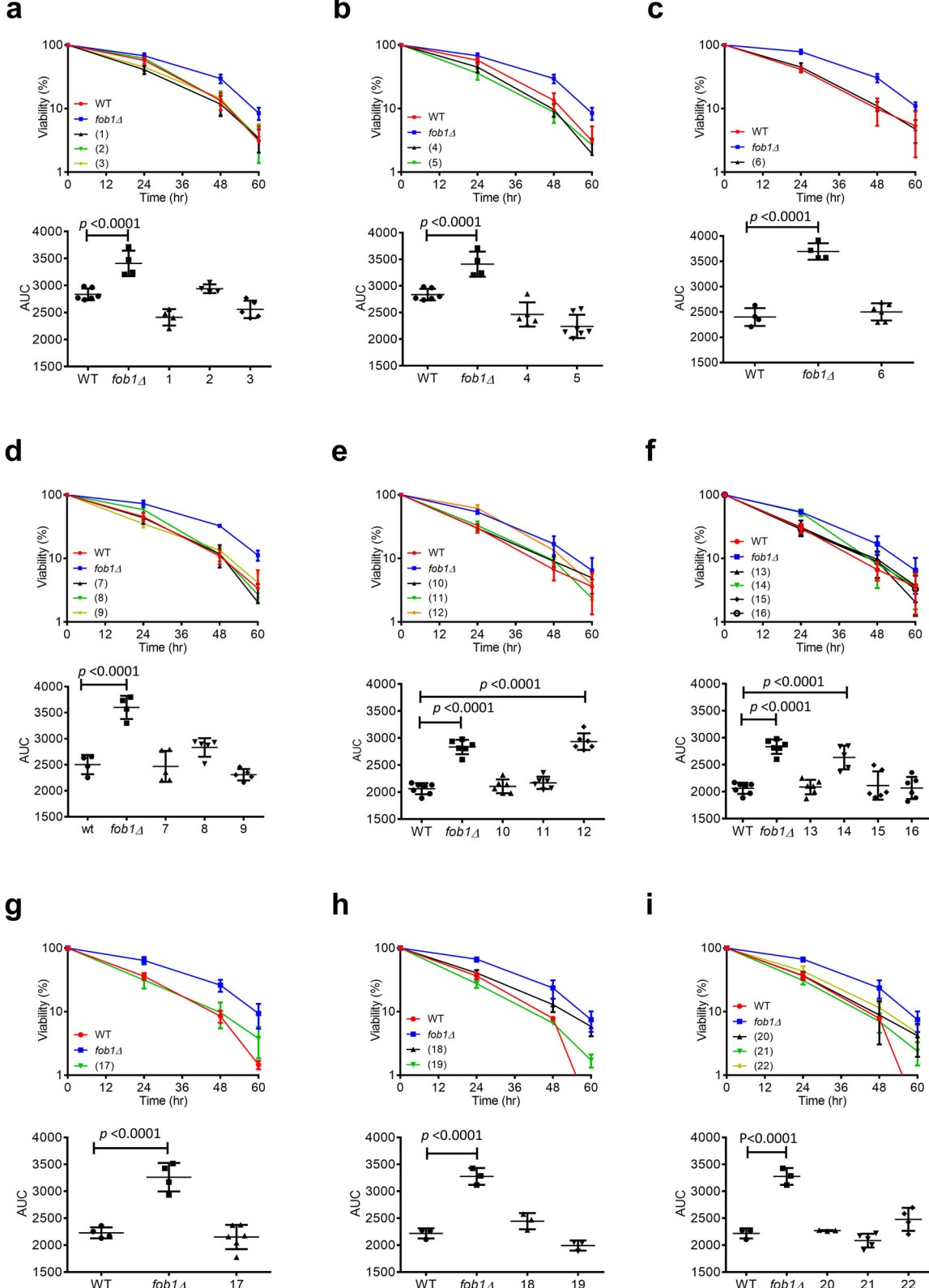

a Sir2-dependent or Tor1-dependent pathway by a micro-manipulation assay. As the data show, corylin promoted RLS in the *sir2Δfob1Δ* double deletion strain (Fig. 4b) and failed to increase RLS in the *tor1Δ* deletion strain (Fig. 4c), which is consistent with the data shown in Fig. 1f. This result suggested that corylin extends lifespan through the Tor1-dependent pathway. Tor (mTOR complex in mammals) is a key signaling

response to nutrients and modulates cell growth and metabolism. Additionally, a previous study showed that inhibiting tor1 relocates Msn2/4 from the cytoplasm to the nucleus and promotes Pnc1 expression[22]. Pnc1 converted nicotinamide to nicotinic acid as a precursor in $NAD^+$ salvage, which increases $NAD^+$ levels, activating stress responses and increasing lifespan[23,24]. First, we determined whether corylin relocates Msn2 to the nucleus and

**Fig. 3 MEP assay of the pure compounds from the hexane extract of *Psoralea corylifolia*. a–i** Visualization curve of haploid MEP strain ZHY1 in liquid YEPD containing (1) psoralen ($n = 4$), (2) isopsoralen ($n = 4$), (3) psoralidin ($n = 5$), (4) 2′,3′-dihydro-2′,3′-dihydroxypsoralidin ($n = 5$), (5) dehydroisopsoralidin ($n = 7$), (6) (+)-bakuchiol ($n = 6$), (7) bavachinin ($n = 5$), (8) bavachin ($n = 5$), (9) chromenoflavanone ($n = 5$), (10) daidzen ($n = 6$), (11) neocorylin ($n = 6$), (12) corylin ($n = 6$), (13) corylifol A ($n = 6$), (14) neobavaisoflavone ($n = 5$), (15) 7-hydroxy-3(2-(2-hydroxypropan-2-yl) benzofuran-5-yl)-4*H*-chromen-4-one ($n = 6$), (16) (2″*S*)-2″,3″-dihydroxyisopentyl-daidzein ($n = 6$), (17) 6-prenyl-fisetin ($n = 6$), (18) 3″-hydroxy-isopentyl-isoliquiritigenin ($n = 3$), (19) isobavachromene ($n = 3$), (20) psorachromene ($n = 3$), (21) 4-methoxybavachalcone ($n = 5$), and (22) isobavachalcone ($n = 4$). Cultures were incubated at 30 °C for 60 h. The viability is presented as CFUs per 500 µl, and this value was determined by harvesting samples at the indicated time points. Data presented as mean ± SD from at least three independent experiments. *p* values were determined by one-way ANOVA (multiple comparisons). Source data are provided as a Source data file.

further increases Pnc1 levels in yeast. Fluorescence microscopy showed that corylin relocated Msn2 in the nucleus to form foci, similar to what was seen in two conditions of calorie restriction (Fig. 4d, e). We next utilized the Pnc1-GFP strain to observe Pnc1 expression by fluorescence microscopy and Western blot analysis. Fluorescence microscopy showed that corylin significantly increased Pnc1 in a dose-dependent manner (Fig. 4f), and the Pnc1 protein expression level was significantly upregulated by corylin treatment (Fig. 4g). $NAD^+$ could be indirectly promoted by Pnc1, and several studies have shown that increases in $NAD^+$ itself ameliorate aging-associated diseases[25]. Thus, we assessed whether corylin elevates $NAD^+$ levels. As shown in Fig. 4h, $NAD^+$ levels were significantly increased following corylin treatment. Caloric restriction (CR) is a well-known strategy for extending lifespan by reducing tor1 signaling[26]. CR modifies the rate of metabolism; reduces the age-associated accumulation of oxidatively damaged proteins, lipids, and DNA; regulates gene expression; and delays aging[2]. Given the evidence linking tor1 inhibition and CR, we next investigated whether corylin increases yeast RLS by a pathway mimicking CR. In yeast, CR was performed by reducing glucose from 2 to 0.5%, and we found that CR increases RLS, as previously reported[27]. Strikingly, corylin failed to extend the RLS under CR conditions (Fig. 4i). In addition, we determined whether corylin influences the CLS (chronological lifespan). Although no significant difference was observed in the overall populations with or without corylin treatment, corylin increased the CLS at the late stage (Supplementary Fig. 5).

**Corylin increases yeast RLS by targeting the Gtr1 protein.** Since corylin increases RLS in a Tor1-dependent manner, we tested possible targets involved in the Tor pathway in yeast by using molecular docking software, and the docking results were displayed by Discover Studio based on the references. By the docking analysis, we found that Gtr1 is a potential target of corylin (Fig. 5a). Furthermore, the docking analysis demonstrated that three different domains may interact with corylin, and one of the interaction domains which contain Ile166 and Trp167 showed stronger interaction with corylin (Fig. 5b and supplementary Fig. 6). To confirm the interactions between Gtr1 and corylin, [1]H NMR chemical shift experiments were conducted (Supplementary Fig. S7). Four different peptides were designed as probes to investigate the interaction between Gtr1 and corylin. (i) peptide 1: WT sequence, (ii) peptide 2: Ile166 and Trp167 were substituted with glutamine, (iii) peptide 3: Ile166 and Trp167 were omitted from the sequence, and (iv) peptide 4: Ile166 and Trp167 were relocated in the sequence (Fig. 5c). Meanwhile, we found that Gtr1 (Rag A in mammals) is highly conserved across species, especially the corylin binding region that we proposed (Fig. 5d). As shown in Fig. 5e, peptide 1 showed strong signal shifts at 8.13, 8.00, 7.94, 7.90, 7.86, and 7.80 ppm upon corylin treatment. Interestingly, peptide 2 and peptide 3, which lacked Ile166 and Trp167, showed no signal shifts in the presence of corylin. In addition, peptide 4, in which Ile166 and Trp167 were relocated, showed weak signal shifts at 8.10 and 7.70 ppm. This result

suggested that the residues and locations of Ile166 and Trp167 might be important for the interaction of corylin. Moreover, we performed a control with daidzen, which shares the major skeleton with corylin and did not increase the RLS in the MEP assay. We found that peptide 1 showed weak signal shifts upon daidzen treatment and strong signal shifts upon corylin treatment in the [1]H NMR spectrum (Supplementary Fig. 8). This result indicates that the aromatic rings on both corylin and daidzein could cause chemical shift perturbations, and it also indicates that peptide 1 interacts more specifically with corylin than daidzein since the signal shifts are much stronger in the presence of corylin than in the presence of daidzen (Fig. 5e and Supplementary Fig. 8). To determine whether corylin targeting Gtr1 to inactivate TOR1 signaling results in lifespan extension, the *gtr1Δ* deletion and Gtr1 overexpression strains were subjected to micromanipulation. The RLS of the *gtr1Δ* deletion strain was increased compared to that of the WT, and *gtr1Δ* combined with corylin did not further increase the RLS in yeast (Fig. 5f). Furthermore, corylin was unable to increase the RLS at higher concentrations in the WT and *gtr1Δ* strains (Fig. 5g). As shown in Fig. 5h, the pGAL-GTR1 strain overexpressed GTR1 protein in YEPG and impaired its expression in YEPD. In YEPD, the pGAL-GTR1 strain increased RLS, and corylin failed to further increase RLS (Fig. 5i). Moreover, we found that WT slightly increased RLS in the YEPG. Most importantly, the overexpression of Gtr1 decreased RLS, while the overexpression of Gtr1 counteracted the lifespan extension property of corylin (Fig. 5j). In conclusion, these data suggested that corylin extends RLS by blocking Gtr1 activates in yeast. In addition, we determined whether corylin increases MSN2 translocation in *gtr1Δ* strain. Using fluorescence microscopy, we found out that the *gtr1Δ* strain exhibited abundant MSN2 foci in the nucleus, and corylin fail to further increase nuclear MSN2 accumulation in this strain (Supplementary Fig. 9).

**Corylin ameliorates cellular senescence in HUVECs.** To extend our knowledge of corylin on a mammalian system, we next investigated whether corylin alleviates the senescence process in HUVECs. In mammalian cells, p21 and SA-β-gal are replicative exhaustion signatures[28]. As shown in Fig. 6a, corylin increased population doubling (PDL). In the late stage of HUVECs, the p21 expression level increased without corylin, indicating cell cycle arrest. However, the p21 expression level decreased under the corylin treatment (Fig. 6b, c). Additionally, corylin decreased SA-β-gal-positive senescent cells compared with the untreated group (Fig. 6d, e). To understand the mechanism by which corylin ameliorates cellular senescence in HUVECs, we next used RNA sequencing to analyze transcriptome differences in three different groups: proliferating cells (Y), senescent cells (S), and senescent cells with corylin treatment (S + C). First, we compared the S:Y (S/Y), S + C:S (S + C/S), and S + C:Y (S + C/Y) groups to verify the transcriptome changes between them. Next, we focused on 433 shared transcripts between S/Y and S + C/S (Fig. 6f). By comparing the KEGG pathway annotation, we found shared

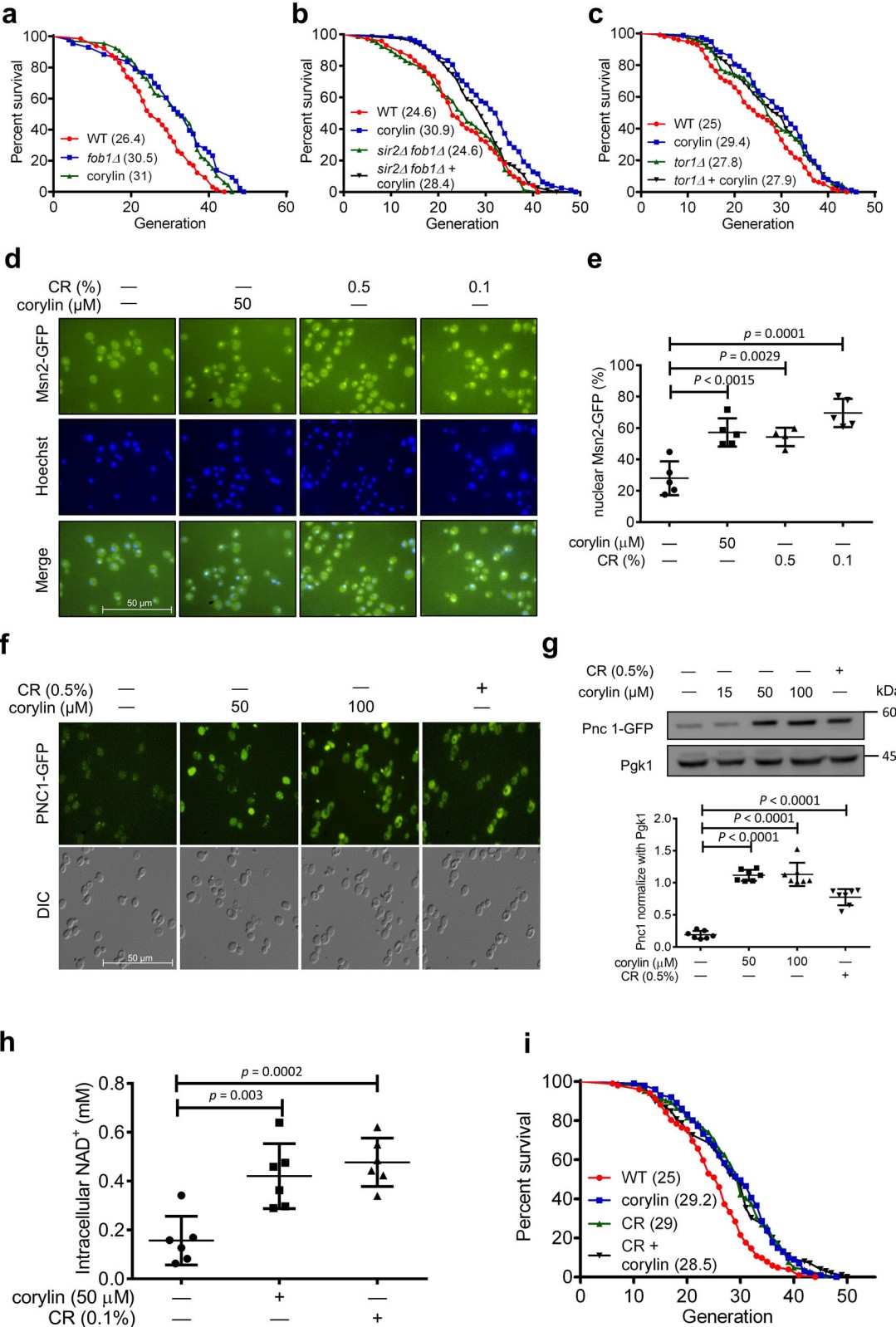

pathways between the two groups associated with cellular senescence, the cell cycle, DNA replication, and the p53 signaling pathway (Fig. 6g). We next decoded the transcriptome differences in those pathways. Numerous transcripts were shifted in S/Y, such as a reduction in cyclin protein, replication factor, DNA helicase, cell proliferation signals, and an increase in SASP, indicating that HUVECs are in the senescent state (Fig. 6h).

Furthermore, we found that the transcriptome pattern of senescent cells treated with corylin was similar to that of young cells (Fig. 6i). Most importantly, we found that eukaryotic translation initiation factor 4E (eIF4E)-binding protein 1 (4E-BP1) down-regulated and Rho upregulated with corylin treatment, which are the downstream mTOR1 and mTOR2 pathways, respectively. Additionally, we found that corylin decreased mTOR1 and

**Fig. 4 Corylin extends the replicative lifespan by regulating Tor1 in *Saccharomyces cerevisiae*.** RLS were determined by micromanipulating individual yeast cells on YEPD with or without corylin. **a** The mean for WT = 26.4 generations ($n = 65$); *fob1Δ* = 30.5 generations ($n = 43$); 15 μM corylin = 31 generations ($n = 66$). **b** The mean for WT = 24.6 generations ($n = 72$); 15 μM corylin = 30.9 generations ($n = 78$); *sir2Δ fob1Δ* double deletion (*sir2Δ fob1Δ*) = 24.6 generations ($n = 78$); *sir2Δ fob1Δ* double deletion with corylin 15 μM (*sir2Δ fob1Δ+* corylin) = 28.4 generations ($n = 75$). **c** The mean for WT = 25 generations ($n = 97$); 15 μM corylin = 29.4 generations ($n = 77$); *tor1Δ* deletion = 27.8 generations ($n = 104$); *tor1Δ* deletion plus 15 μM corylin (*torΔ+* corylin) = 27.9 generations ($n = 100$) from at least three biologically independent experiments. **d** Msn2-GFP cells were treated with various concentrations of corylin and caloric restriction conditions (CR; 0.5 and 0.1%) for 1 h. cells were treated with various concentrations of corylin and CR conditions (0.1%) for 1 h. **e** Quantification of MSN2 ($n = 5$ biologically independent experiments). **f** Pnc1-GFP cells was performed by immunofluorescence microscopy. **g** Immunoblotting was performed using anti-GFP and anti-PGK1 antibodies The Pnc1-GFP expression level was normalized to PGK1 ($n = 7$ biologically independent experiments). **h** Intracellular NAD+ concentration was detected using the BY4741 strain with or without corylin and the CR condition (0.1%) for 2 h ($n = 6$ biologically independent experiments). **i** RLS were determined by micromanipulating individual yeast cells on YEPD with or without corylin. The mean for WT = 25 generations ($n = 102$); 15 μM corylin = 29.2 generations ($n = 101$); CR = 29 generations ($n = 104$); CR with 15 μM corylin (CR + corylin) = 28.5 generations ($n = 99$). Data presented as mean ± SD. **a–c, i**, *p* values were determined by the Gehan–Breslow–Wilcoxon test (see also Supplementary Fig. 14); **e, g, h**, *p* values were determined by two-tailed unpaired Student's *t*-test. Source data are provided as a Source data file.

p70S6k phosphorylation in U2OS cells (Supplementary Fig. 10). Taken together, these data reveal that corylin ameliorates cellular senescence.

**Corylin prolongs lifespan in aged obese mice.** We next asked whether corylin increases the C57BL/6J lifespan under metabolic stress. To determine the long-term effect of corylin in C57BL/6J mice, forty-week-old mice in this study were fed a HFD or HFD plus 0.1% (w/w) corylin (HFD/C) ad libitum for the remainder of their lives. Four weeks after corylin supplement (at 44 weeks of age), the survival curves of the HFD and HFD/C groups began to diverge and remained separated till the end of experiments. By 102 weeks of age, 63.3% of the HFD-fed control mice had died, compared with 43.3% of the HFD/C-fed mice. Corylin supplement significantly increased the lifespan of aged mice fed on high fat diet (Fig. 7a). Notably, the maximum difference in the survival rate between the mice supplied with corylin and the HFD-fed mice reached 30.0% at 92 weeks. Over the course of the experiment, the bodyweight trajectories and food intake did not differ between the two groups (Fig. 7b, c), suggesting that the beneficial lifespan is exerted by corylin instead of by less caloric intake. To investigate the pharmacokinetics of corylin in mice, we monitored the circulating level of corylin after oral gavage. The daily dose of corylin was calculated as 50 mg per kg bodyweight based on the food intake of mice was approximately 2 g of HFD plus corylin (0.1% w/w) per day in mice with an average bodyweight of 40 g. The average serum level of corylin reached 0.68 μM at 1 h after corylin oral gavage and remained at 0.26 μM at 15 h (Fig. 7d). These data suggested that corylin supplement benefits to longevity in aged mice under metabolic stress. To investigate the beneficial effect of corylin supplement in age-associated functional and metabolic dysregulations. We first tested rearing behavior, including vertical activity and behavior, to evaluate corylin effect on age-related physical dysfunction in aged HFD-fed mice. Motor coordination was examined by rotarod tests, which revealed physical function of muscle strength and balance that is impaired by aging. The rearing behavior showed that HFD/C-fed mice exhibited strongly increased activity compared with HFD-fed mice (Fig. 7e). In the rotarod test, HFD/C-fed mice had better motor skills as they aged than HFD-fed mice (Fig. 7f, g). This result indicated that the age-related decline in terms of physical function is ameliorated by corylin supplement. Next, we verified whether corylin inhibited the phosphorylation of mTor1 in the *gastrocnemius muscle*, which is involved with mobility and muscle strength[29]. Immunoblotting showed that corylin impairs the mTor1 phosphorylation in the *gastrocnemius muscle* which consists with our hypothesis (Fig. 7h, i). In addition to decline in physical function, the risk of aging-associated pathology is increased during aging that reflects on multiple metabolic parameters. At the end of the experiment

(102 weeks of age), we collected and analyzed fasting serum parameters in both groups of aged male mice. The fasting blood glucose, total cholesterol, low-density lipoprotein (LDL), and tri-glyceride (TG) levels were reduced in HFD/C-fed mice (Fig. 7j, k). The corylin-mediated reduction in serum lipid parameters may ameliorate the risk factors for metabolic syndrome during aging, which benefits to overall health of aging. Next, we assessed the effects of corylin on tissue functional markers. In this analysis, lower levels of a hepatic damage marker, aspartate amino-transferase (AST), and of a marker of renal function, creatinine, were found in HFD/C-fed aged mice compared to HFD-fed aged mice, indicating that corylin supplement prevents aging-associated organ functional decline (Fig. 7k). Thus, we concluded that corylin extends lifespan by protecting against age-associated and obesity-associated metabolic and functional declines in aged mice. This finding indicates that daily corylin supplement could benefit to overall functionality during aging process and therefore improves quality of life.

**Discussion**

CR influences a wide range of fundamental processes to promote lifespan in organisms[30]. The concept of Tor signal inhibition by CR is demonstrated[31]. Yet, the hypothesis that Sir2 contributes to CR is still controversial. In our study, corylin significantly increased the RLS of yeast in both WT and *sir2Δfob1Δ* deletion strains indicates corylin increased lifespan by a sir2-independent pathway (Fig. 4a, b). However, lifespan extending ability of corylin in *sir2Δfob1Δ* deletion strain is less than in WT strain (20% vs. 10%). Therefore, we suggested that inhibition of Tor1 by corylin could activate multiple signaling pathways to increase lifespan. In turn, when Tor signaling is inhibited, Sir2 is one of the downstream mediators that extends the lifespan of yeast. In agreement with other studies, we first hypothesized that the inhibition of Tor1 could indirectly activate Sir2. Numerous studies have strongly linked Pnc1 expression to Sir2 activity[32]. Our study demonstrated corylin suppresses TOR1 activity and upregulates Pnc1 expression in yeast. This indicates that tor1 could also be a upstream regulator of sir2[22]. Second, we hypothesized the inhibition of Tor in the Sir2-deficient strain would activate other pathways to compensate for Sir2 function, increasing the lifespan of yeast. Sir2 plays an important role in yeast lifespan by suppressing rDNA recombination. The *sir2Δ* deletion strain shows deregulation of rDNA recombination, resulting in rapid accumulation of ERCs and a decreased lifespan of yeast[33]. However, a previous study showed that in sir2-deficient yeast, tor1 inhibition activated the MSN2/4 or sir2 homolog gene (Hst1-4) to increase lifespan by suppressing rDNA recombination in yeast[22,34]. Based on a previous study, Sir2 activity is subordinate to Tor1 inhibition. Without Sir2, the inhibition of Tor signaling

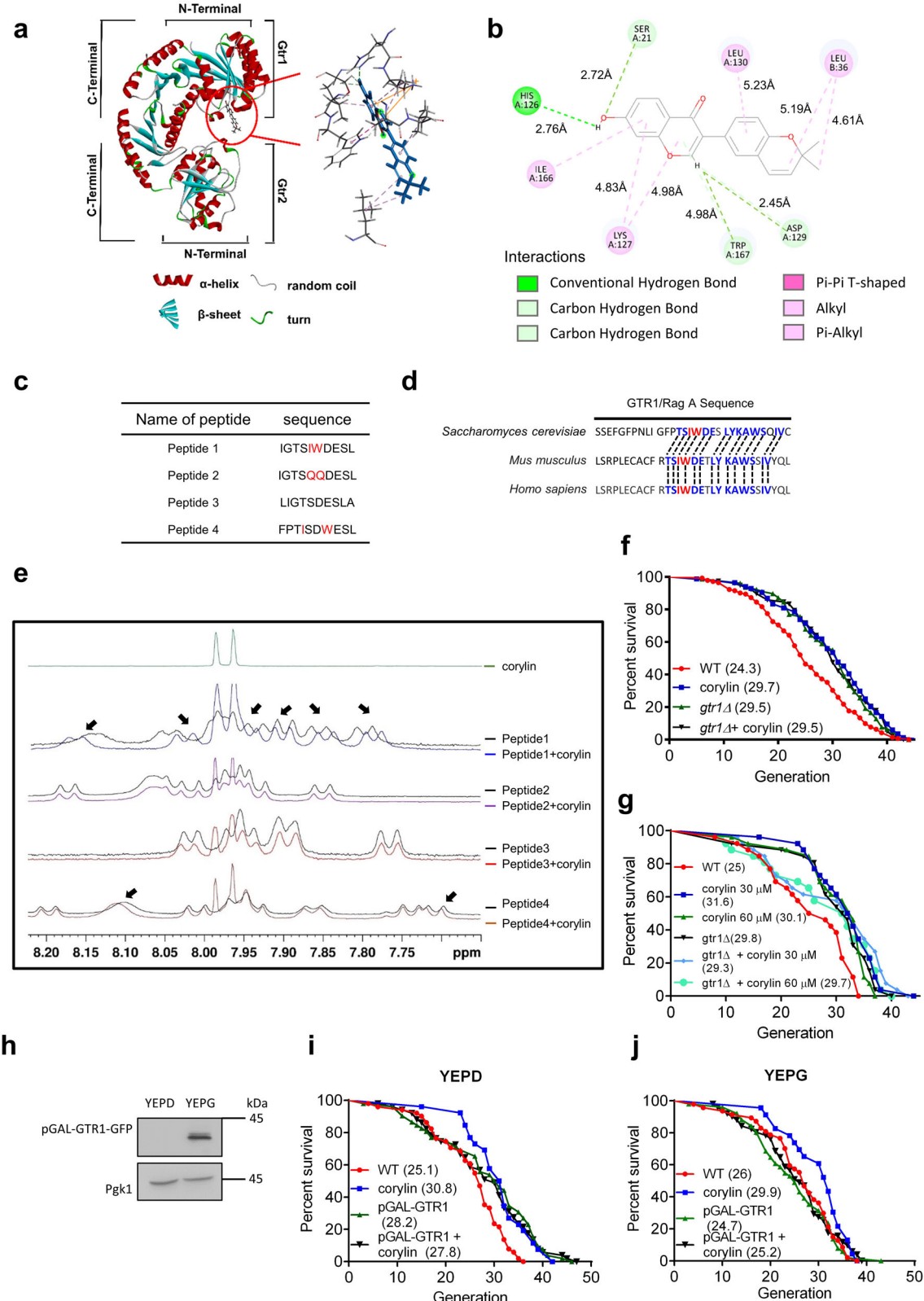

activates multiple pathways to increase the lifespan of yeast. This finding may also justify our conclusion that corylin extends lifespan in a Sir2-independent manner.

In the past two decades, TOR signaling has been widely discussed regarding its signal reduction contributes to lifespan increase. In addition, the EGO complex has a conserved function as a regulator of TOR1[35]. Extensive studies have shown that in the EGO complex, the Gtr1-Gtr2 complex activates TOR1 and initiates anabolic signaling. Furthermore, the loss of Gtr1 consequently decreases Sch9 phosphorylation[36]. These results indicate that Gtr1 is an important regulator of TOR1 signaling. However, the limited study has shown a potential increase in RLS by reducing Gtr1 signaling with a loose criterion, indicated by a discrepancy between a-type and α-type yeast[37]. Here, we directly

**Fig. 5 Corylin docked with Gtr1 to inactivate the TOR1 pathway. a** A docking model of corylin in the N-terminal domain of Gtr1 shown as a ribbon (PDB ID: 3R7W). Ligand–protein interactions with the binding residues of Gtr1 (light gray) and corylin (blue). **b** Ligand–protein interactions with the binding residues of Gtr1 and corylin. The green dashed lines indicate hydrogen bonds and the pink dashed lines indicate π-interactions. **c** The peptide sequence for the chemical shift assay. **d** The Gtr1/Rag A sequence 159–180 in various species. **e** $^1$H NMR spectrum of peptide in the presence or absence of corylin. The arrows indicate the upfield chemical shift in the presence of corylin at a molar ratio of corylin and peptide of 1:1. **f** The RLS was determined by micromanipulating individual yeast cells on YEPD with or without corylin. The mean for WT = 24.3 generations ($n = 77$); 15 μM corylin = 29.7 generations ($n = 75$); $gtr1\Delta$ = 29.5 generations ($n = 78$); and gtr1Δ with 15 μM corylin (gtr1Δ+ corylin) = 29.5 generations ($n = 78$). **g** The mean for WT = 25 generations ($n = 26$); 30 μM corylin = 31.6 generations ($n = 26$); 60 μM corylin = 30.1 generations ($n = 26$); gtr1Δ = 29.8 generations ($n = 26$); gtr1Δ+ 30 μM corylin = 29.3 generations ($n = 26$); gtr1Δ+ 60 μM corylin = 29.7 generations ($n = 26$). **h** Immunoblotting was performed using anti-GFP and anti-PGK1 antibodies, and the pGAL-Gtr1 strain was cultured in YEPD and YEPG. **i** The RLS was determined by micromanipulating individual pGAL-Gtr1 yeast cells on YEPD with or without corylin. The mean for WT = 25.1 generations ($n = 51$); 15 μM corylin = 30.8 generations ($n = 52$); pGAL-GTR1 = 28.2 generations ($n = 52$); and pGAL-Gtr1 with 15 μM corylin (pGAL-Gtr1+ corylin) = 27.8 generations ($n = 51$). **j** pGAL-Gtr1 yeast cells were cultured on YEPG to conduct a micromanipulation assay. The mean for WT = 26 generations ($n = 47$); 15 μM corylin = 29.9 generations ($n = 46$); pGAL-Gtr1 = 24.7 generations ($n = 51$); and pGAL-Gtr1 with 15 μM corylin (pGAL-Gtr1 + corylin) = 25.2 generations ($n = 51$) from at least three biologically independent experiments. **f**, **g**, **i**, **j**, $p$ values were determined by the Gehan–Breslow–Wilcoxon test (see also Supplementary Fig. 14). Source data are provided as a Source data file.

show that the lifespan of the *gtr1Δ* deletion strain can be increased (Fig. 5f). Most importantly, we demonstrated that corylin targets the Gtr1 protein, which inactivates TOR1 signaling to increase RLS.

Cellular senescence is characterized by loss of DNA repair capacity, chromosome instability, and telomere erosion[38]. In our RNA sequencing data based on corylin treatment, histone cluster family genes and double-strand break response genes were shown to be upregulated compared with senescent cells. The cyclin-dependent inhibitor p21 contributes to cell cycle arrest. Long noncoding RNA LINC00899 and homologous recombination protein BRCA1 negatively regulate p21 expression[39,40]. We found upregulation of LINC00899 and BRCA1 in our RNA sequencing data. Furthermore, telomerase reverse transcriptase (TERT) and KLF4, the transcription regulator of TERT, were upregulated compared with the senescence group, indicating the good integrity of telomeres[41]. Whether these pathways are related to mTOR inhibition or whether corylin has other potential targets contributing to a slower senescence process require further investigation. However, our results have clearly demonstrated the potential of corylin to ameliorate cellular senescence.

In a previous study, a HFD induced metabolic stress and accelerated senescent cell accumulation to further increase the risk of death[18]. In our animal experiment showing that corylin extends lifespan in mice fed with HFD, corylin increased survival by 20% at 102 weeks. Of note, the reduction in the risk of death was better than that produced by resveratrol in obese mice[42]. Rapamycin, an mTOR1 inhibitor, has been shown to increase the lifespan and decrease weight gain in a HFD model[43,44]. However, rapamycin elicited hyperglycemic effects via mTOR2 inhibition with long-term treatment, which could be reversed by rapamycin discontinuation[45,46]. This indicates that universal inhibition of mTOR signaling could generate side effects[47]. Although the bodyweight was similar to that of the HFD group following corylin treatment, the fasting blood glucose level was lower than that of the HFD group. Furthermore, we found that corylin decreased TGs in HFD-fed mice, which was not observed with rapamycin[48]. Taken together, these results suggested that inhibiting/partly inhibiting mTOR1 signaling by Rag A but not mTOR2 could be an attractive intervention for increasing lifespan. Despite extensive studies in the field of aging, the lack of systematic methods to screen and validate lifespan extension compounds lead to very few effective compounds were identified. Here, we extract knowledge from TCM pharmacopeia, and exploited MEP to efficiently and scientifically identify lifespan-extending TCMs in yeast. Most importantly, we are the first group to report that *P. corylifolia* and one of its major constituents, corylin, extend lifespan in various models. Moreover, the molecular mechanisms of aging are highly related to aging diseases. Consistent to our findings, our previous study revealed that corylin inhibits vascular cell inflammation, proliferation, migration, and reduces atherosclerosis in ApoE-deficient mice, indicating that corylin has the potential to treat aging-associated diseases[49]. We previously showed that corylin ameliorates obesity by activating adipocyte browning and mitigating insulin resistance in diet-induced obesity (DIO) mice[50]. In addition, corylin reduced ORO and hepatic fibrosis in the short term of HFD suggesting that corylin protects against hepatic steatosis in mice with HFD (Supplementary Fig. 11).

In conclusion, we have proposed an efficient method for screening and validating beneficial compounds to extend lifespan, with strong evidence showing the lifespan extension properties of corylin in yeast and cell models and improving health and survival by facilitating stress resistance in HFD mice.

## Methods

**Ethical statement.** All animal procedures complied with all relevant ethical regulations and were conducted under approved protocols (Protocol# CGU15-150) by Chang Guan University, Research Guidelines for the Care and Use of Laboratory Animals.

**Genotypes of yeast strains.** To overexpress GTR1, we used a plasmid generated by Longtine et al. that replaced the GTR1 promoter with a regulated promoter by galactose with a GFP tag[51]. The Genotypes of yeast strains in this article are shown in Table 2.

**General experimental procedure.** Optical rotations of compounds were measured on a JASCO P-1010 polarimeter using a 10 cm cell. UV spectra were recorded on a Hitachi UV-2010 spectrophotometer, and IR spectra were recorded on a JASCO FT-IR-4000 spectrometer. NMR spectra were recorded on a Bruker AVANCE-400. Proton and carbon NMR spectra were measured on a 400 MHz instrument. Mass spectrometry data were obtained on a Finnigan TSQ 700 mass spectrometer. HPLC separations were performed on a HITACHI L-2130 HPLC equipped with a HITACHI L-2400 refractive index detector.

**Chemicals.** Peptone was obtained from OXOID (LP0037; Basingstoke, UK), and yeast extract was obtained from BD (212750; Sparks, MD, USA). DMSO (D8418), estradiol (E2758-1G) and ADH (A7001-15KU) were obtained from Sigma-Aldrich (St. Louis, MO, USA), and TCA was obtained from Nippon Shiyaku Kogyo (Osaka, Japan).

**TCM materials.** All of the TCMs were supplied and authenticated by the Chuang Song Zong pharmaceutical company (Pingtung, Taiwan). A voucher specimen (CGU-PC-1) was deposited in the herbarium of Chang Gung University, Taoyuan, Taiwan.

**Extraction and isolation of *P. corylifolia*.** The dried fruits (5.4 kg) of *P. corylifolia* were repeatedly extracted with ethanol (11 l × 4) and extracted five times with ethanol at 70 °C for 4 h (11 l × 5). The combined crude extracts (1.4 kg) were partitioned sequentially between $H_2O$ (842.38 g) and *n*-hexane (557.62 g). The *n*-

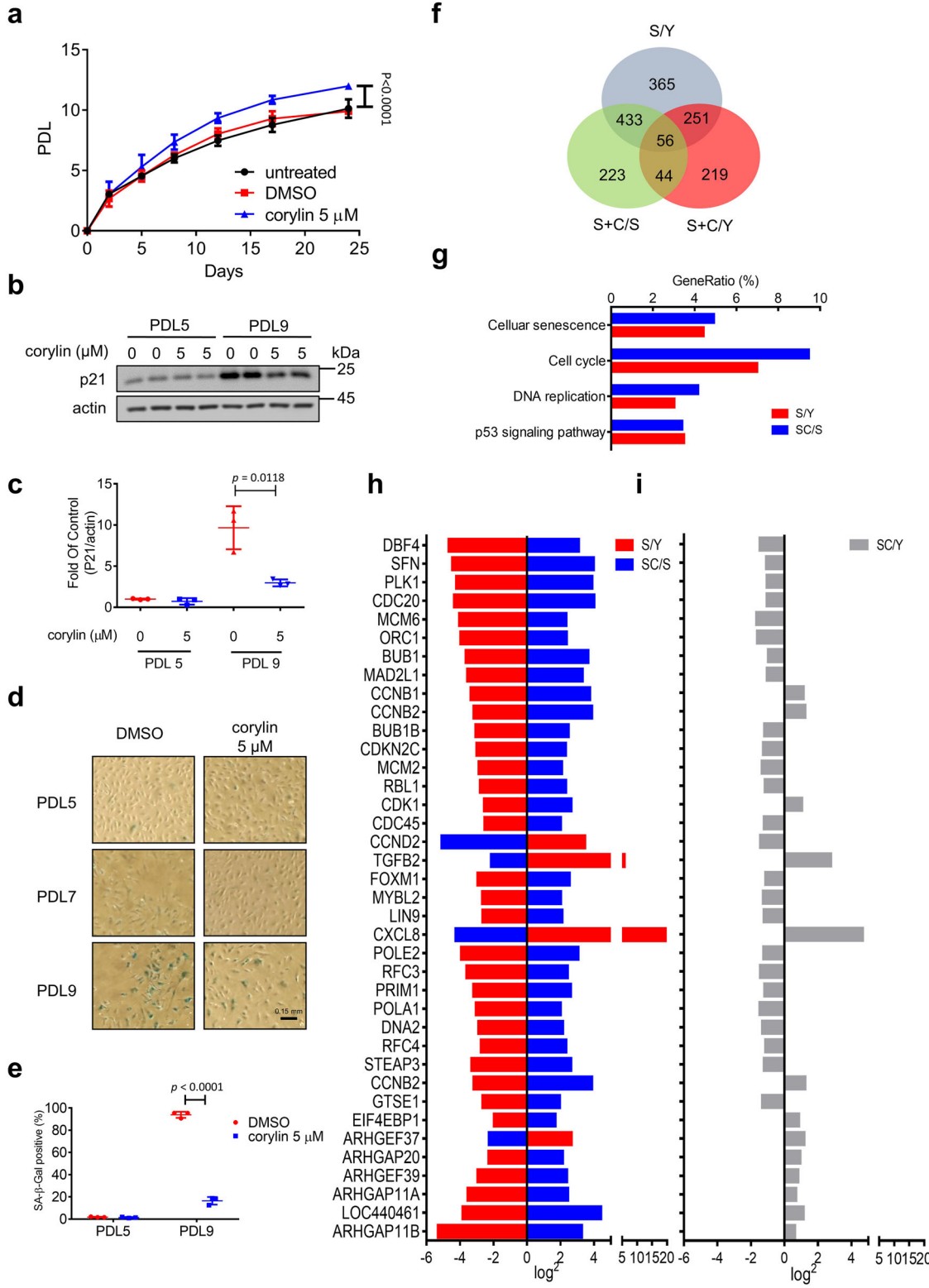

hexane layer was separated by silica gel chromatography using sequential mixtures of *n*-hexane and ethyl acetate. The 20:1 = H:E mixture eluted compound 6; the 10:1 = H:E mixture eluted (in order of elution) compound 1, 2, 7, and 18; the 7:1 = H:E mixture eluted (in order of elution) compound 5, 8, 9, 11, 13, and 22; the 5:1 = H:E mixture eluted (in order of elution) compound 12, 14, 17, 19, 20, and 21; and the 1:1 = H:E mixture eluted (in order of elution) compound 3, 4, 10, 15, and 16. The extraction scheme of *P. corylifolia* was presented in Supplementary Fig. 12 and the corylin NMR proton spectra were presented in Supplementary Fig. 13.

**Mother enrichment program**. The liquid aging assay was performed as previously described[12]. Briefly, cells were cultured at 30 °C in YEPD medium overnight, and the cultures were diluted to $A_{600} = 0.25$ and recovered to the log phase. Cultures were counted and inoculated in culture tubes at a cell density of $2 \times 10^3$ cells/ml. The mixtures contained 17 β-estradiol at a final concentration of 1 μM, and they were incubated in a roller drum at 50 rpm and 30 °C for 60 h. Each group was monitored by harvesting samples at the indicated time points. The collected samples were washed and plated in solid media, and the viability is reported as CFUs per 500 μl. We used the OpenCFU 3.8 software to calculate colonies.

**Fig. 6 Corylin alleviates cellular senescence in HUVECs. a** Measurement of the proliferative capacity of cells cultured with or without corylin (5 µM). Cells were passaged at regular intervals and counted. Cell numbers were used to establish a growth curve, displaying cumulative population doublings. The population doubling level (PDL) of human umbilical vein endothelial cells (HUVECs) was assessed ($n = 3$ biologically independent experiments). **b**, **c** Cell lysates were prepared from different PDLs of HUVECs that were cultured with or without corylin, and immunoblotting was performed using anti-p21. Quantification was normalized with actin ($n = 3$ biologically independent experiments). **d** SA-β-GAL staining was performed at PDL 5, 7, and 9 to detect senescent cells (scale bar = 0.15 mm). **e** Quantification of senescence was performed by calculating the ratio of SA-β-gal-positive cells. At least 100 cells were calculated per group in the experiment ($n = 3$ biologically independent experiments). **f** The possible overlapping and nonoverlapping transcripts in three comparison groups. **g** KEGG was enriched in the shared senescence signatures between S/Y and SC/S (Y young cells, S senescence cells, SC senescence cells + corylin). **h** The shared senescence signatures between S/Y and SC/S. **i** SC/Y, involving the pathway from top to bottom, Senescence marker; cell cycle; DNA replication; p53 signaling pathway. Data presented as mean ± SD from at least 3 biologically independent experiments. **a** $p$ values were determined by two-way ANOVA (multiple comparisons); **c**, **e** $p$ values were determined by two-tailed unpaired Student's $t$-test. Source data are provided as a Source data file.

**Micromanipulation assay**. The micromanipulation of the yeast cells was carried out as described previously[52]. Prior to analysis, strains were plated onto fresh solid medium and grown for 2 days at 30 °C. Single colonies were grown in YEPD medium overnight at 30 °C, and a small number of cells were then plated onto a fresh YPD plate for lifespan analysis. After overnight growth on the lifespan plates, the cells were arrayed on the plate using a micromanipulator and allowed to grow for approximately 3 h. Virgin daughter cells were selected and subjected to lifespan analysis. In the lifespan experiments, the plates were incubated at 30 °C during the day and stored at 4 °C overnight. Each experiment consisted of more than 70 mother cells and was independently repeated three times.

**Immunoblotting**. Yeast protein was prepared as previously described[53]. The samples were loaded in 10% SDS-polyacrylamide gel. The following primary antibodies were used for immunoblotting at a 1:1000 dilution: anti-Pgk1 (ab113687; Abcam, UK), anti-GFP (G1544; Sigma-Aldrich; St. Louis), anti-P21 (#2947; cell signaling, USA), anit-p-mTOR (#2971, cell signaling, USA), anti-T-mTOR (#2972, cell signaling, USA), anti-β-actin for cell (A5441, Sigma-Aldrich, USA), and anti-β-actin for mice tissue (GTX109628, GeneTex, USA). Secondary antibodies were obtained from Sigma-Aldrich (St. Louis, MO, USA) and used at a 1:100,000 dilutions. Images were acquired with a Wealtec KETA-CL imaging system.

**Intracellular NAD$^+$ content**. Cells were harvested ($4 \times 10^8$ per group), washed with 50% DMSO and water, pelleted and stored at −80 °C. The extractions were performed with 250 µl of 1 M formic acid saturated with butanol. After incubation for 30 min on ice, 62.5 µl of 100% TCA (W/V) was added to each sample, and the samples were then incubated on ice for an additional 15 min. The samples were pelleted by centrifugation at $17,000 \times g$ for 5 min, the supernatant was transferred to another Eppendorf tube, the pellets were washed with 125 µl of 20% TCA, and the material was repelleted by centrifugation. The combined supernatants were used for the following tests. For analysis, each sample was assembled in reaction buffer (100 µl of extract, 400 µl of 360 mM Tris, 240 mM lysine, pH 9.7, 0.24% (v/v) EtOH; the control group had 5 µl of water, and the ADH group had 5 µl of 5 mg/ml alcohol dehydrogenase). After 5 min at room temperature, the absorbance of each sample was measured at 340 nm. The NAD$^+$ content of the cells in each sample was determined relative to the water group (as the basal NADH level) and the alcohol ADH catalytic group.

**Fluorescence microscopy**. Yeast cultures were pre-grown overnight in YEPD medium. Then, yeast cells were recovered twice to the log phase in YEPD. For the Msn2 relocation assay, Msn2-GFP cells were cultured in 5 ml of culture liquid with or without corylin in a flask for 1 h. A total of $2 \times 10^7$ cells were harvested, and samples were washed with 100 mM HEPES and stained with Hoechst (#33342; Sigma-Aldrich; St. Louis, MO, USA) containing 3.7% formaldehyde for 5 min at room temperature. The samples were washed twice with HEPES and spotted onto slides for observation. Nuclei were stained with Hoechst #33342 to identify live cells. The number of cells with nuclear Msn2-GFP was counted manually and normalized to those with Hoechst staining. For the Pnc1-GFP strain, $2 \times 10^7$ cells were harvested and washed twice with YEP. Each group was collected for further analysis with a Nikon ECLIPSE Ni-U plus fluorescence microscope equipped with a 100× oil objective. Images were acquired with a DS-U3 CCD camera that was controlled with NIS-Element BR 4.0 software.

**Molecular docking analysis**. The crystal structure of yeast Gtr1–Gtr2 was used for analysis (PBD ID: 3R7W). The docking analysis was performed using BIOVIA Discovery Studio v19.1.0.18287. The protein was prepared, and the ligand was minimized before docking. Docking was performed using the standard protocol.

**Chemical shift experiment**. NMR binding experiments were carried out with peptide substrates on a 400 MHz Bruker DRX spectrometer equipped with a BBFO

probe at 25 °C. Data sets were the average of 100 scans. All NMR spectra were collected in the presence of peptide, corylin, and 99.9% DMSO-d6. The $^1$H, $^{13}$C, DEPT135, COSY, HSQC, HMBC, and ROESY spectra of peptide 1 by 600-MHz Bruker spectrometer equipped with a TXI cryoprobe at 25 °C (Supplementary Fig. 15). Moreover, the synthesis report of peptide 1 from the outsourcing company is included in Supplementary Data Fig. 15.

**Cell culture**. HUVECs were originally obtained from Bioresource Collection and Research Center, Taiwan. HUVECs were cultured in M199 supplemented with HEPES, ECGS (Millipore), heparin, NaHCO$_3$, L-glutamine, sodium pyruvate, and FBS (20% final concentration). The cells were grown in an atmosphere of 5% CO$_2$ at 37 °C and sub-cultured by trypsinization with trypsin-EDTA (Lonza). Cells were seeded at $8 \times 10^4$ in 3.5 cm culture dishes and passaged such that the monolayers never exceeded 90% confluency. A sample was collected in every passage for cell extraction, RNA extraction and SA-β-gal staining. The cells were propagated until senescence, and cell numbers were determined when sub-cultured. Population doublings (PDs) were estimated using the following equation: PDL = 3.32 (log (total viable cells at harvest/total viable cells at seeding).

**SA-β-gal staining**. HUVECs were treated with or without corylin. Then, when the cells reached 90% confluency, they were washed twice with phosphate-buffered saline (PBS) and fixed for 5 min with 2% formaldehyde and 0.2% glutaraldehyde. The cells were then incubated at 37 °C for 18 h with a staining solution (40 mmol/l citric acid, sodium phosphate, pH 6.0, 1 mg/ml 5-bromo-4-chloro-3-isolyl-β-D-galactoside (X-gal, Sigma), 5 mmol/l potassium ferrocyanide, 5 mmol/l potassium ferricyanide, 150 mmol/l NaCl, and 2 mmol/l MgCl$_2$). Each group was collected for further analysis with an Olympus $1 \times 71$ microscope equipped with a 10× objective. Images were acquired with a PCO. Panda 4.2 camera that was controlled with cameware64 software. Senescence-associated (SA)-β-gal-positive cells were observed by microscopy, and over 300 cells were counted in at least three independent fields.

**RNA sequencing**. RNA was collected from HUVECs and extracted by TRIzol (T9424, Sigma, USA). The extracted RNA samples were sent to Genomics (Taipei, Taiwan) for analysis, and a library was constructed. Briefly, after quality control of raw reads, mRNA was purified using reverse transcriptase and a random primer to synthesize single-strand cDNA, and dUTP was used in place of dTTP to generate double-stranded cDNA. A single "A" nucleotide was added to the 3′ end of ds cDNAs. Then, multiple indexing adapters were ligated to the 5′ and 3′ ends of ds cDNA. PCR was used to selectively amplify the DNA fragments with adapters on both ends. The library was validated on an Agilent 2100 Bioanalyzer and Real-Time PCR System. The gene ratio and expression change were summarized by Kyoto Encyclopedia of Gene and Genomes (KEGG) pathway data and Gene Ontology ($p$-value < 0.05, log-2 > 2).

**Animals**. All animal procedures complied with all relevant ethical regulations and were conducted under approved protocols (Protocol# CGU15-150) by Chang Guan University, Research Guidelines for the Care and Use of Laboratory Animals. C57BL/6J male mice (34 weeks of age) were provided by the National Laboratory Animal Center (NLAC), NAR Labs, Taiwan. All mice were housed in individual cages and maintained at room temperature at 23 ± 1 °C and humidity (45–65%) with a 12 h dark/light cycle. After reaching 40 weeks of age, the mice were randomly divided into two groups: Group I (HFD), fed a HFD with 54% fat ($n = 30$), and Group II (HFD/C), fed a HFD containing corylin (1 g corylin/1 kg HFD) ($n = 30$). Corylin (purity ≥ 98%) was purchased from Shanghai BS Bio-Tech Co., Ltd., China. The composition of the test diets is shown in Table 3.

**Pharmacokinetic analysis**. C57BL/6J male mice of blood samples were collected at 1, 3, 6, 9, 12 and 15 h after corylin oral gavage, and serum samples were then separated. Serum samples (100 µl) were mixed with ice-cold acetonitrile (150 µl) at

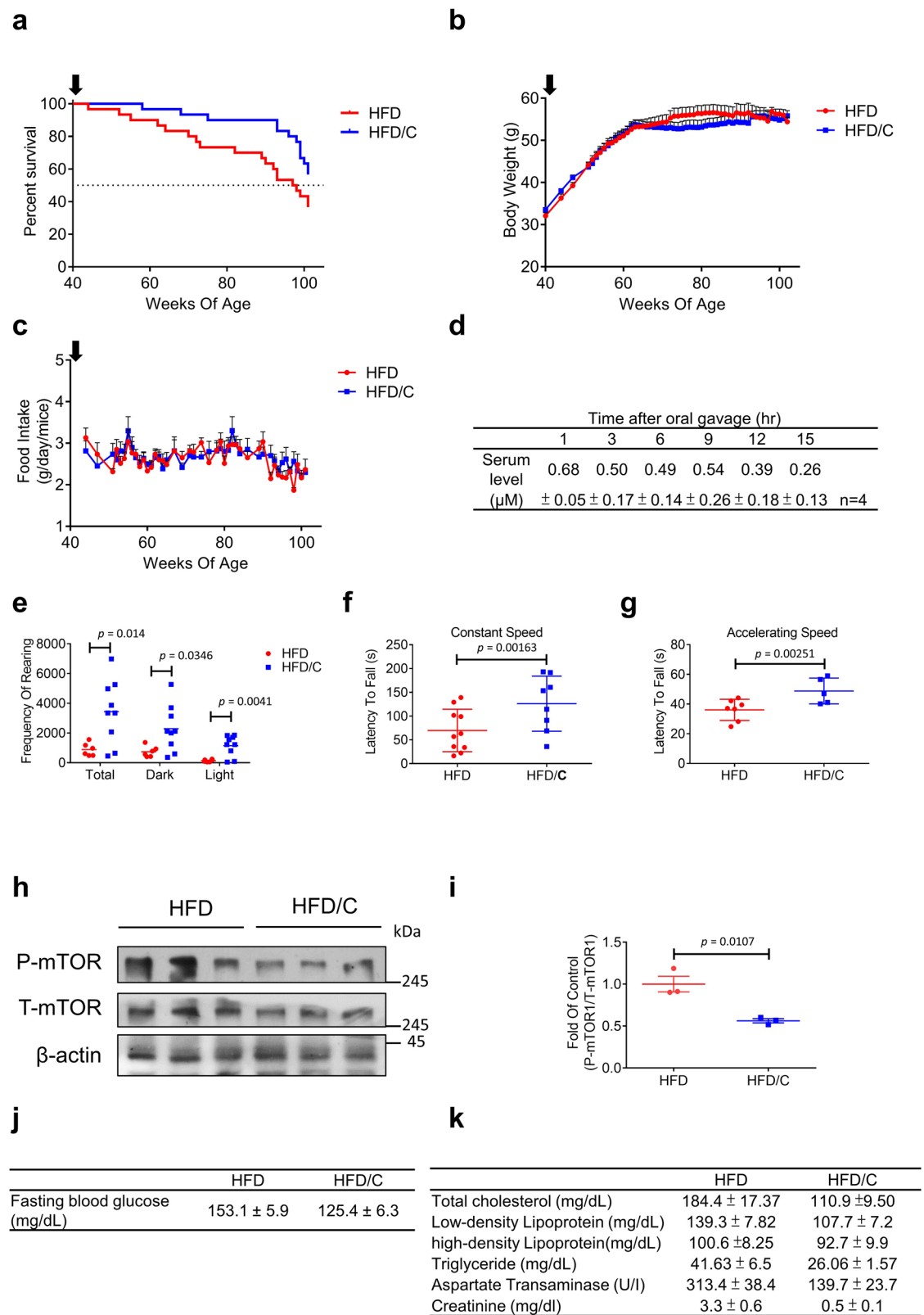

4 °C for 1 h and then centrifuged for 15 min at 15,000 × g and 4 °C. After centrifugation, the supernatant (150 µl) was harvested in a new eppendorf tube, and 150 µl of ddH₂O was added. The sample was analyzed by an LC/FTMS system.

**Fasting blood glucose level**. To determent the blood glucose, the C57BL/6J male mice at 102 weeks of age were fasting for 16 h. The blood samples were obtained by cutting the tip of tail and the blood glucose was measured by ACCU-CHEK (Roche).

**The rotarod test**. The C57BL/6J male mice were tested on a rotarod at 81 weeks of age. The mice were habituated for 3 days, during which they were placed on the rotarod at a constant speed (4 rpm) and had to remain on the rotarod for 1 min every day. Testing on each day consisted of five trials with a 10 min rest between each trial. The acceleration trial began with the rotarod set at an initial rate of 4 rpm, accelerating to a maximum of 40 rpm within 5 min. The constant trial began with the rotarod set at a rate of 4 rpm and lasted for a maximum of 200 s. The latency to falling was recorded, and the average latency to falling was calculated for each trial.

**Fig. 7 Corylin prolongs lifespan in aged HFD-fed mice.** Forty-week-old male mice were fed a HFD with or without corylin treatment ($n = 30$ per group). **a** Survival curves of mice fed a HFD versus 1%(v/v) corylin (HFD/C) show a significant (Gehan–Breslow–Wilcoxon test $\chi^2 = 4.887$ and $p = 0.0271$) improving lifespan. **b, c** The bodyweight of mice after started fed with HFD or HFD/C and food consumption (The start point to fed corylin indicated by a black arrow, $n = 30$ at start point). **d** Circulating level of corylin ($n = 4$ biologically independent samples). **e** Rearing behavior was analyzed in obese mice fed a HFD with or without corylin for 35 weeks under dim and bright light conditions (HFD: $n = 6$; HFD/C $n = 9$ biologically independent animals). Latency to falling in the **f** constant rotarod and (HFD: $n = 10$; HFD/C: $n = 8$ biologically independent animals) **g** accelerating rotarod tests (HFD: $n = 7$; HFD/C: $n = 5$ biologically independent animals). **h** Western of total muscle for Phospho-mTOR Ser2448 (P-mTOR), total-mTOR (T-mTOR) and β-Actin ($n = 3$). **i** Quantification was normalized with T-mTOR ($n = 3$ biologically independent samples). **j** Fasting blood glucose level result ($n = 9$ biologically independent samples, $p = 0.0058$). **k** Serum biochemical marker results ($n = 5$ biologically independent samples, total cholesterol $p = 0.0037$, low-density lipoprotein $p = 0.0151$, triglyceride $p = 0.0486$, aspartate transaminase $p = 0.0006$, creatinine $p < 0.0001$). Values are expressed as the mean ± SEM. **a** $p$ values were determined by the Gehan–Breslow–Wilcoxon test. **e–g, i–k** $p$ values were determined by two-tailed unpaired Student's $t$-test. Source data are provided as a source data file.

**Table 2 Genotypes of yeast strains.**

| Strain | Genotype | Source |
|---|---|---|
| BY4741 | MATa his3Δ leu2Δ met15Δ ura3Δ | This study |
| ZHY02 | MAT alpha ade2::HisG his3 leu2 met15Δ::ADE2 trp1Δ63 hoΔ::SCW11pr-CRE-EBD78-NatMX loxP-UBC9-loxP-LEU2 loxP-CDC20-Intron-loxP-HPHMX | Tyler lab[54] |
| BJY02 | MAT alpha ade2:: His G his3 leu2 met15Δ::ADE2 trp1Δ63 hoΔ::SCW11pr-CRE-EBD78-NatMX loxP-UBC9-loxP-LEU2 loxP-CDC20-Intron-loxP-HPHMX fob1::KAN | Tyler lab[54] |
| DCY05 | MATa his3Δ leu2Δ met15Δ ura3Δ GFP-PNC1 | This study |
| gtr1Δ | MATa his3Δ leu2Δ met15Δ ura3Δ gtr1::KAN | Taiwan Yeast Bioresource Center |
| HSY002 (pGAL-GTR1) | MATa his3Δ leu2Δ met15Δ ura3Δ pGal-GFP-GTR1::KAN | This study |

**Table 3 Composition of the test diets.**

| | HFD | HFD/C | |
|---|---|---|---|
| | g/kg diet | | Source |
| Casein | 254 | 254 | Sigma |
| Cellulose | 61 | 61 | Sigma |
| Sucrose | 321 | 321 | General stores |
| Soybean oil | 10 | 10 | General stores |
| Unsalted butter | 290 | 290 | General stores |
| AIN-93G Mineral mixture | 44.5 | 44.5 | MP Biomedicals |
| Ain-93 Vitamin mixture | 12.5 | 12.5 | Dyets, Inc. |
| L-Cystine | 4 | 4 | Sigma |
| Choline bitartrate | 3 | 3 | Sigma |
| Corylin | – | 1 | Shanghai BS Bio-Tech |
| kcal/g | 5.016 | | |
| CHO calorie/total calories (%) | 25.7 | | |
| Fat calorie/total calories (%) | 54.0 | | |
| Protein calorie/total calories (%) | 20.3 | | |

**Rearing behavior.** Rearing behavior was analyzed in C57BL/6J male mice fed a HFD with or without corylin for 35 weeks (75 weeks of age) using the OxyletPro System. Before the metabolic rate was monitored, the mice were individually caged for 24 h to acclimate to the system.

**Serum parameter analysis.** The blood of C57BL/6J male mice for serum biochemistry was collected before sacrifice (102 weeks of age). The serum LDL, TG, and total cholesterol levels were measured using specific reagent kits (Fortress Diagnostics, Antrim, Northern Ireland). The serum glucose and high-density lipoprotein (HDL) levels were measured using specific reagent kits (Randox, Antrim UK). Insulin was measured by ELISA (Mercodia, Uppsala, Sweden). HOMA-IR was calculated as the fasting insulin level (mU/l) × blood glucose level (mmol/l)/22.5. The serum AST and CK levels were measured using a Fuji Dri-Chem 4000i analyzer (Fujifilm, Tokyo, Japan).

**Statistics and reproducibility.** Graphic visualization and statistical analyses were performed by GraphPad Prism 6. All values are provided as the mean ± SD. The exact $p$ value was provided in figures or legend. For the MEP assay, fluorescence microscopy, western blotting, SA-β-gal staining, rotarod test, and rearing behavior experiment,

significance was determined via Student's $t$-test. For the HUVECs PLD, significance was determined by two-way ANOVA (multiple comparisons). At least three independent replicates were performed for each experiment. For the survival curve of the mouse model, significance was determined via the Gehan–Breslow–Wilcoxon test.

**Reporting summary.** Further information on research design is available in the Nature Research Reporting Summary linked to this article.

## Data availability

The data generated in this study are provided in the Supplementary Information/Source Data files. Source data are provided with this paper. Further information will be available from the corresponding author on reasonable request (chinchuan@mail.cgu.edu.tw). The RNA-sequence data generated in this study have been deposited in the Gene Expression Omnibus under the accession GSE196828. The data are also available in the Harvard Dataverse under accession code 7ERYZN. Structure that support the findings of this study was downloaded from the PDB database (https://www.rcsb.org/structure/3R7W) Source data are provided with this paper.

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

## Acknowledgements
We are grateful for the generous gift of yeast strains from Jessica Tyler and Dan Gottschling. We thank the staff of the Taiwan Yeast Bioresource Center at the First Core Labs, National Taiwan University College of Medicine, for bioresource sharing. The NMR spectra were carried out at the Metabolomics Core Laboratory, Healthy Aging Research Center (HARC), Chang Gung University. HARC was funded by Ministry of Education in Taiwan (MOE) (EMRPD1L0421). This work was supported by grants from Chang Gung Memorial Hospital (CMRPD1K0381 to C.-C.C.; CMRPD1K0641 to Y.-L.L.) and the Ministry of Science and Technology (109-2628-B-182-007 to C.-C.C.; 110-2628-B-182-016 to C.-C.C.) of Taiwan.

## Author contributions
Conceived and designed the experiments: C.-C.C., S.-H.W., Y.-L.L., W.-C.L., and Y.-C.C.; performed the experiments: W.-C.T., T.-H.W., C.-H.K., C.-Y.C., and S.-F.C.; analyzed the data: C.-C.C., C.-C.W., Y.-L.L., C.-H.F., S.-L.Y., J.-J.S., S.-H.Y., and C.-J.W.; contributed reagents/materials/analysis tools: C.-Y.C. and T.-L.H.; wrote the paper: C.-C.C., W.-C.T., and C.-Y.L.

## Competing interests
The authors declare no competing interests.
