## [Peer Review File · Nature Communications]

The flavonoid corylin exhibits lifespan extension properties in mouseREVIEWER COMMENTS

Reviewer #1 (Remarks to the Author):

Overall, the authors of this manuscript delved into the knowledge of traditional Chinese medicine to try to identify bioactive components that could potentially lead to lifespan extension. The initial studies are comprehensively conducted in yeast systems and the authors also try to examine how the identified a compound of interest (corylin, a flavonoid) that alters biological responses in HUVEC cells and mice fed a high fat diet. While the manuscript contains suitable information for publication, a few points must be addressed before this manuscript can be considered for publication.

1. Selection of TCM single herbs based on use to treat aging diseases? What are these aging-diseases? How does one classify a condition as aging-related and non-aging-related? This approach can likely lose a lot of actually good candidates.
2. Use of herbal formulas – you don't know what specific/pure ingredient in the formula is giving rise to the biological/anti-aging effects. Therefore, the authors must clarify the exact criteria that was used to identify the bioactive ingredients in these formulations.
3. Figure 3 – how were the concentrations chosen/determined for the different compounds? It does state that the data is not shown. The authors should provide this information on how they were all subjected to various concentrations in the MEP program.
4. Please provide evidence for the translational relevance of the MEP to mammalian systems. Did the authors test the more traditional replicative and chronological lifespan assessment in yeast?
5. The authors show that corylin (the major bioactive component identified through this study) targets Gtr1-Tor. However, there are a number of other studies showing that corylin also targets other pathways (Yu et al. FASEB Journal 2020). Are these mechanisms dose-dependent or are they all part of the same cellular signaling networks that are selectively activated by corylin?
6. The authors specify in line 297 that "...these data were consistent with our hypothesis that corylin suppresses the mTOR pathway increases lifespan". This statement is not backed by evidence. This is shown to some degree in the yeast system but in the HUVECs in question, only transcriptome changes that hint at correlative changes as opposed to causal changes were assessed. Moreover, direct evidence linking mTOR are not shown. The authors should include these data in the manuscript or modify writing in this section to reflect the data presented.
7. Similarly, what do the authors mean by "...the transcriptomes all showed a certain level of improvement in S+C/S"? What is an improvement in this sense? Depending on the context, the proliferative capacity induced by corylin could be detrimental (ie. are these cells becoming less senescent but more cancerous?). Therefore, it is dangerous to make such claims without fully acknowledging the contexts are consequences.
8. In the mouse study, body weight trajectories are not presented. These must be presented to accurately understand how these mice were performing metabolically.
9. Details about how many mice were used for the HFD-lifespan study must be readily available in the figure legend. Also, the start of the treatment regimen must be shown with an arrow on the lifespan curve and food consumption graphs.
10. It seems like the lifespan study was concluded before both groups reached the 50% survival mark. The authors should provide justification for this. Were tissues collected from these mice? Could these be used for further study of organ-specific effects of corylin? Is the mTOR inhibitory properties of corylin seen in yeast seen in HFD mice as well? (Ma et al. Hypertension Research 2010). In the absence of these data, the mammalian aspects of the work cannot be fully integrated into the overall message of the manuscript.
11. Please provide details of the C57BL6 substrain us or origin ("J", "N", "CRL"), and the sex used. What is the effect of the compound in standard chow fed animals?
12. What was the cause of death and the pathological findings at necropsy? Did the authors performed any sacrifices during the lifespan experiment to gain insights on protection on hepatic steatosis?

Reviewer #2 (Remarks to the Author):

Wang et al. screened many different extracts for lifespan extension using the mother enrichment program in *Saccharomyces cerevisiae*. A single extract and subsequently corylin, a single molecule identified within the extract was found to extend yeast replicative lifespan (RLS). Corylin did not further extend the RLS of a *tor1* or *gtr1* mutant. Corylin treatment resulted in increased nuclear localization of Msn2-GFP and increased fluorescence and steady state levels of Pnc1-GFP. Corylin also increased intracellular NAD⁺ levels. Docking studies suggested corylin may bind to Gtr1, a GTPase that simulates the TOR complex 1 (TORC1). Changes in the NMR shift of a small peptide based on part of the expected corylin Gtr1 binding site supported a possible interaction of corylin with this region of Gtr1. Additionally, the authors test corylin in HUVEC cell culture and observe indications that corylin reduced senescence due to continued culturing of the cells. Mice fed a high fat diet (HFD) were treated with corylin and the shortened lifespan was extended, while food intake remained the same. The corylin treated mice also had reduced fasting glucose, reduced cholesterol levels, and reduced AST and creatine levels at 102 weeks old compared to untreated mice. The HFD fed mice treated with corylin also had increased rearing behavior, and improved performance on rotorod tests.

Major comments:

It seems the binding of corylin to Gtr1 would be one of the more interesting aspects of this manuscript. The identification of novel small molecule inhibitors of proteins tends to lead to novel insight into biological processes. However, the NMR data of a small peptide potentially interacting with corylin offers only a small amount of support for this hypothesis, but it may just suggest that tryptophan and/or isoleucine may be able to interact with corylin. The epistasis lifespan experiments provide an additional piece of support. Further *in vivo* support of the hypothesis that Corylin binds to Gtr1 would be meaningful. For example, it would be relatively trivial for the authors to make a *gtr1* MSN2-GFP or *gtr1* PNC1-GFP strain where the effects of corylin on Msn2 localization or Pnc1 abundance would be expected to be prevented. This experiment would also support the authors claims of Corylin mediated Tor1 dependent effects on Msn2/4. Additionally, if corylin has any effects on growth rate at high concentration, the *gtr1* null mutant would be expected to be resistant and the assay would be easy to perform.

The types of statistical analysis do not appear to be indicated in the paper and it is critical that these be included in methods and the figure legends.

Additional comments:

The first sentences of the abstract "In long history of traditional Chinese medicine (TCM), some single herb and complex formulas have been recorded to increase lifespan in TCM pharmacopeia. However, the mechanism of these TCMs increasing lifespan is insufficient."

This reads like a factual statement and it seems likely that the lifespan extension claim is not based on scientific evidence. Stating that the mechanism is insufficient further suggests that it is. Similarly, the Table 1 title "Lists of TCM single herbs and TCM herbal formulas used to treat aging diseases and/or extend lifespan" as stated may imply that modern evidence exists for the claim.

Line 64-65 "Despite such extensive studies in the field of aging, the number of compounds that extend lifespan is very small". There are numerous compounds that have been published to extend the lifespan of various model organisms, perhaps this sentence should be worded something like "the number of compounds identified by the interventions testing program that extend the lifespan of mice is very small" or something like rigorously identified to extend lifespan in mice. A similar statement is made at the beginning of the results section (line 124: "only a few compounds have been shown to increase lifespan").

Lines 70-72, "Of the available screening methods for lifespan-extending drugs, the SIRT1 *in vitro* assay is the most common method, as it allows relatively simple and fast data collection". I don't think this is the most common method of screening for lifespan-extending drugs, perhaps more appropriate would be "one method" instead of "most common method".

In the results section, it should be indicated that the other 77 candidates presumably did not show evidence for lifespan extension under the conditions tested. The concentrations tested and extraction method for the other extracts could perhaps be noted in the event other researchers

found the information useful.

Lines 212-214 "Additionally, several studies have shown that inhibiting tor1 relocates Msn2/4 from the cytoplasm to the nucleus, and promotes PNC1 expression [22]." The sentence states "Several studies", yet only one is referenced.

"Pnc1 hydrolyzes nicotinamide to nicotinic acid as a precursor in NAD⁺ salvage, which increases NAD⁺ levels, activating stress responses and increasing lifespan".

Has Pnc1 been shown to increase lifespan? The reference is a review that references a paper simply showing the deletion of PNC1 prevents lifespan extension by dietary restriction (glucose reduction).

Msn2/4 localizes to the nucleus in response to many different stressors and the relation to Tor1 in Figure 4 is inferred but is not directly supported by any evidence. Increased Msn2-GFP nuclear localization being due to corylin regulation of Tor1 seems like an overly strong conclusion based on the data in this figure. Given Pnc1 is a Msn2/4 target, this only supports the observed effect with Msn2/4. The experiments proposed above would yield more direct support.

Lines 226-28 "As shown in Figure 4H, NAD⁺ levels were significantly increased following corylin treatment, suggesting that tor1 signaling was inhibited by corylin treatment." Does tor1 deletion increase NAD⁺ levels? This correlation would seem to be necessary for the argument and would likely be known and could be referenced.

How was nuclear localization of Msn2-GFP scored? E.g., manually or some threshold for puncta? Should be included in methods.

What was the vehicle for corylin and what was the control condition used (equal concentration of vehicle only?). Generally it is advisable to be sure the vehicle has no effect by including an additional control with untreated (no vehicle).

Figure 1. The legend indicates a "10 µg/ml ethanol extract of *P.corylifolia* or DMSO", but the figure itself indicates ethanol extract of "*Fructus Psoraleae*". This seems an odd error.

n-HEAXEN instead of n-hexane is included in panels 1D,E, and F.

Figure 6a, was a no DMSO control also performed?

Figure 5b and 5c, The 0 and 5 indications are slightly ambiguous. It appears to indicate the µM concentration of corylin, but that should be more clearly indicated in the legend for the panel. It may be easier for a reader to interpret G and H if "S/Y" and "S+C/S" were defined in the legend.

Interestingly, the mice food intake was unchanged. Rapamycin (an mTOR inhibitor) reduced food intake in HFD C57BL/6 mice (Geng-Tuei et al. J. Pharm. Sci 2009). This would suggest differences in when intervening against Gtr1 vs mTorc1 (assuming similar HFD was performed). Is it known that rapamycin or mtorc1 knockdown has similar effects to corylin on glucose and cholesterol in mice fed HFD? Comparisons of mtor1 knockouts and rapamycin with HFD from literature here would be useful.

For the NMR data, were coupling constants determined? Wouldn't the corylin aromatic peaks in the spectrum also be expected to shift? Rather than remove the two residues thought to be important for the peptide binding to corylin and add two residues at the N and C terminal ends, the substitution of the residues would have seemed like a better control peptide. And perhaps a peptide with tryptophan and isoleucine at different locations may be meaningful in terms of the specificity of the interaction. It is being assumed that the small peptide is taking on a structure similar to that observed within the folded protein, but the simplicity of using a peptide to model a part of the folded protein makes interpretation of the results difficult.

The discussion ends "with strong evidence showing the lifespan extension properties of corylin in multiple organisms". One could argue that normal lifespan extension was only shown in yeast, and the HFD data in mice shows that corylin provides some stress resistance in this organism.

-Brian Wasko

Black: our response

Underlined: changed in our manuscript. In the revised manuscript, we also highlighted changes in red.

Dear Editor and Reviewers,

We are submitting a revised version of our manuscript (NCOMMS-20-48816-T), entitled “Validation of bioactive components from traditional Chinese medicine for lifespan extension”.

We thank the reviewers for their extremely careful and constructive reviews of our work. We have provided point-by-point responses to the reviewers’ specific comments below.

Reviewer #1 (Remarks to the Author):

Overall, the authors of this manuscript delved into the knowledge of traditional Chinese medicine to try to identify bioactive components that could potentially lead to lifespan extension. The initial studies are comprehensively conducted in yeast systems and the authors also try to examine how the identified a compound of interest (corylin, a flavonoid) that alters biological responses in HUVEC cells and mice fed a high fat diet. While the manuscript contains suitable information for publication, a few points must be addressed before this manuscript can be considered for publication.

1. Selection of TCM single herbs based on use to treat aging diseases? What are these aging-diseases? How does one classify a condition as aging-related and non-aging-related? This approach can likely lose a lot of actually good candidates.

Response:

The majority of TCM candidates are compounds listed in pharmacopeias that promote health and longevity, and some of them are from the National Health Insurance Research Database (NHIRD) and are used to treat age-related diseases such as osteoporosis, sarcopenia, and dementia.

To clarify, we have changed the sentence to “To choose TCM candidates for validation, we searched the National Health Insurance Research Database (NHIRD) as well as several pharmacopeias, such as the Compendium of Materia Medica, Qian jin yao fang, Shennong Materia Medica, and Huangdi neijing, for TCMs that may increase lifespan and are used to treat age-associated diseases, for example, osteoporosis, sarcopenia, and dementia.”

The reviewer is correct; this approach can likely omit many good candidates. The study was aimed to identify an efficient and systemic method for screening and validating TCMs that extend the lifespan. By using the MEP, we provide an

alternative and simple screening method to assess the RLS in yeast and then extend the work to cell and mouse models. Currently, we are very excited and confident that the system is working; therefore, additional TCM candidates are now in screening.

2. Use of herbal formulas – you don't know what specific/pure ingredient in the formula is giving rise to the biological/anti-aging effects. Therefore, the authors must clarify the exact criteria that was used to identify the bioactive ingredients in these formulations.

Response:

We appreciate the reviewer's thoughtful suggestions. The standard protocol for TCM formula extraction was added to the supplementary data (Supplementary Figure. 1) to clarify the exact criteria used for TCMs.

3. Figure 3 – how were the concentrations chosen/determined for the different compounds? It does state that the data is not shown. The authors should provide this information on how they were all subjected to various concentrations in the MEP program.

Response:

In our experience, the effective dose was usually somewhere between 10 and 30 μM . Because some of the components that we obtained from the crude extract of *P. corylifolia* were limited, 15 μM was the maximum concentration that contained all of the compounds needed to perform the MEP assay. Moreover, chalcones 18-22 showed cytotoxicity at 15 μM ; hence, we reduced the dose to 1.5 μM . As requested, some descriptions have been added to the results. "To identify the active compound that increases lifespan in *P. corylifolia*, 22 components were tested using the MEP assay (Fig. 3a-i). Due to the limited amounts of compounds, a concentration of 15 μM was chosen for the MEP assay. The flavonoid subgroup chalcones (18-22) showed cytotoxicity at 15 μM (Supplementary Figure. 4); therefore, the concentration was reduced to 1.5 μM ".

4. Please provide evidence for the translational relevance of the MEP to mammalian systems. Did the authors test the more traditional replicative and chronological lifespan assessment in yeast?

Response:

The MEP assay offers a readily available screening method to examine the genetic

and biochemical processes of yeast replicative aging, which allows us to rapidly verify potential candidates for increasing the RLS. Furthermore, the gene/protein associated with the RLS in yeast was reported to affect the lifespan in mammals (i.e., Tor1/mTorc1; Sir2/SIRT1). (Harrison et al., Nature, 2009; McCormick et al., Cell Metabolism, 2015; Minor et al., Scientific Reports, 2011). Indeed, MEP data still need to be verified by micromanipulation assays for several reasons (Lindstrom & Gottschling, Genetics, 2009). Therefore, we used the MEP assay as a screening tool to identify candidates, which were verified with a micromanipulation assay, and then the results were extended to cell and mammalian models.

As requested, we added new data. We tested the effects of several doses corylin on the CLS. Although there was no significant difference in the whole population with or without corylin treatment, corylin increased the CLS at the late stage (days 30-48). Additional CLS data are now provided in the results. “In addition, we determined whether corylin influences the CLS. Although no significant difference was observed in the overall populations with or without corylin treatment, corylin increased the CLS at the late stage (Supplementary Figure. 5).”

5. The authors show that corylin (the major bioactive component identified through this study) targets Gtr1-Tor. However, there are a number of other studies showing that corylin also targets other pathways (Yu et al. FASEB Journal 2020). Are these mechanisms dose-dependent or are they all part of the same cellular signaling networks that are selectively activated by corylin?

Response:

Different groups (including our own) have reported that corylin is an attractive intervention with anti-obesity and anti-inflammation properties and promotes osteoblast differentiation (Chen et al., Antioxidants (Basel), 2020; Hung et al., Scientific Reports, 2017; Yu et al. FASEB Journal 2020) Interestingly, these signaling pathways are related to longevity (Zhang, Qu, Liu, & Belmonte, Nat Rev Mol Cell Biol, 2020), suggesting that they are likely all part of the same cellular signaling networks of aging, even though corylin seems to target multiple pathways.

6. The authors specify in line 297 that “...these data were consistent with our hypothesis that corylin suppresses the mTOR pathway increases lifespan”. This statement is not backed by evidence. This is shown to some degree in the yeast system but in the HUVECs in question, only transcriptome changes that hint at correlative

changes as opposed to causal changes were assessed. Moreover, direct evidence linking mTOR are not shown. The authors should include these data in the manuscript or modify writing in this section to reflect the data presented.

Response:

We thank the reviewer for asking these questions. As requested, “Taken together, these data were consistent with our hypothesis that corylin suppresses the mTOR pathway and increases lifespan” has now been changed to “Taken together, these data reveal that corylin ameliorates cellular senescence.”

7. Similarly, what do the authors mean by “...the transcriptomes all showed a certain level of improvement in S+C/S”? What is an improvement in this sense? Depending on the context, the proliferative capacity induced by corylin could be detrimental (ie. are these cells becoming less senescent but more cancerous?). Therefore, it is dangerous to make such claims without fully acknowledging the contexts are consequences.

Response:

The reviewer is correct. We thank the reviewer for clarifying this error. We intended to emphasize that the transcriptome pattern of senescent cells treated with corylin was similar to that of young cells. To make this statement less confusing, we added “The shared senescence signatures between SC/Y in Fig. 6I” and changed the sentence to “~~However, the transcriptomes all showed a certain level of improvement in S+C/S. Furthermore, the S+C group showed a similar pattern to that of the Y group (Fig. S1), indicating that with corylin treatment, HUVECs were more proliferative.~~” “Furthermore, we found that the transcriptome pattern of senescent cells treated with corylin was similar to that of young cells (Fig. 6i).”

8. In the mouse study, body weight trajectories are not presented. These must be presented to accurately understand how these mice were performing metabolically.

Response:

We appreciate the reviewer’s thoughtful suggestions. As requested, the body weight trajectories were added to Fig. 7b with elaboration in the context.

9. Details about how many mice were used for the HFD-lifespan study must be readily available in the figure legend. Also, the start of the treatment regimen must be shown with an arrow on the lifespan curve and food consumption graphs.

Response:

As requested, we have now included all of these changes in the figures as well as in the figure legends. The timing of corylin treatment is indicated by a black arrow on the lifespan curve, body weight and food consumption graphs (Fig. 7a-c). Forty-week-old male mice were fed a HFD with or without corylin treatment (n=30 per group).

10. It seems like the lifespan study was concluded before both groups reached the 50% survival mark. The authors should provide justification for this. Were tissues collected from these mice? Could these be used for further study of organ-specific effects of corylin? Is the mTOR inhibitory properties of corylin seen in yeast seen in HFD mice as well? (Ma et al. Hypertension Research 2010). In the absence of these data, the mammalian aspects of the work cannot be fully integrated into the overall message of the manuscript.

Response:

The reviewer is correct; only the HFD group reached the 50% level (median survival 97.5 weeks old); the HFD/C group did not reach this level. For this experiment, we referenced two different groups for the survival pattern and drug treatment duration (Baur et al., Nature, 2006; Zhu et al., Aging cells 2019). In terms of the survival pattern, both HFD groups reached 50%, but the resveratrol- and alogliptin-treated groups did not reach this level. In addition, at the time of sacrifice, the corylin treatment duration was 65 weeks (longer/older than the reference papers), and the survival rate of the HFD with corylin group was still 56.6%, while that of the HFD group was 36.6%. We were concerned that we may not obtain sufficient samples for the HFD group if the HFD/C group reached the 50% level.

We appreciate the reviewer's thoughtful suggestions. As requested, we have added new data and expanded the result. "Next we verified whether corylin inhibited the phosphorylation of mTor1 in the *gastrocnemius muscle* which is involved with mobility and muscle strength. Immunoblotting showed that corylin impair the mTor1 phosphorylation in the *gastrocnemius muscle* which consists with our hypothesis (Fig. 7h-i)."

11. Please provide details of the C57BL6 substrain us or origin ("J", "N", "CRL"), and the sex used. What is the effect of the compound in standard chow fed animals?

Response:

As requested, we have now provided the detail of the substrain and gender "C57BL/6J

male mice” in the material and method.

Indeed, with such a promising result, the effect of the corylin on chow-fed animals immediately raises the question to us. Hence, the effect on chow diet mice under corylin treatment is now processing.

12. What was the cause of death and the pathological findings at necropsy? Did the authors performed any sacrifices during the lifespan experiment to gain insights on protection on hepatic steatosis?

Response:

We did notice that both groups did not show visual injury or abnormally lumps. Mortality seems like a natural cause instead of injury or cancer. However, we regretted that we did not conduct the image documentation.

We agree with the reviewer, whether corylin protect against hepatic steatosis may provide insightful information. Although we did not perform the experiment during the lifespan experiment in mice, we added new data and showed that corylin reduced ORO and hepatic fibrosis in the short term of HFD. Additional details of the data are now provided in the supplemental data and we have now expanded the Discussion “In addition, corylin reduced ORO and hepatic fibrosis in the short term of HFD suggesting that corylin protects against hepatic steatosis in mice with HFD (Supplementary Figure. 9).”

In conclusion, we are extremely grateful for the reviewer’s time, care, and constructive suggestions. We have now incorporated all of the above-mentioned changes in the figures and the text to refine our manuscript, which we feel the manuscript has drastically improved.

Black: our response

Underlined: changed in our manuscript. In the revised manuscript, we also highlighted changes in red.

Dear Editor and Reviewers,

We are submitting a revised version of our manuscript (NCOMMS-20-48816-T), entitled “Validation of bioactive components from traditional Chinese medicine for lifespan extension”.

We thank the reviewers for their extremely careful and constructive reviews of our work. We have provided point-by-point responses to the reviewers’ specific comments below.

Reviewer #2 (Remarks to the Author):

Wang et al. screened many different extracts for lifespan extension using the mother enrichment program in *Saccharomyces cerevisiae*. A single extract and subsequently corylin, a single molecule identified within the extract was found to extend yeast replicative lifespan (RLS). Corylin did not further extend the RLS of a *tor1* or *gtr1* mutant. Corylin treatment resulted in increased nuclear localization of Msn2-GFP and increased fluorescence and steady state levels of Pnc1-GFP. Corylin also increased intracellular NAD⁺ levels. Docking studies suggested corylin may bind to Gtr1, a GTPase that simulates the TOR complex 1 (TORC1). Changes in the NMR shift of a small peptide based on part of the expected corylin Gtr1 binding site supported a possible interaction of corylin with this region of Gtr1. Additionally, the authors test corylin in HUVEC cell culture and observe indications that corylin reduced senescence due to continued culturing of the cells. Mice fed a high fat diet (HFD) were treated with corylin and the shortened lifespan was extended, while food intake remained the same. The corylin treated mice also had reduced fasting glucose, reduced cholesterol levels, and reduced AST and creatine levels at 102 weeks old compared to untreated mice. The HFD fed mice treated with corylin also had increased rearing behavior, and improved performance on rotorod tests.

Major comments:

It seems the binding of corylin to Gtr1 would be one of the more interesting aspects of this manuscript. The identification of novel small molecule inhibitors of proteins tends to lead to novel insight into biological processes. However, the NMR data of a small peptide potentially interacting with corylin offers only a small amount of support for this hypothesis, but it may just suggest that tryptophan and/or isoleucine

may be able to interact with corylin. The epistasis lifespan experiments provide an additional piece of support. Further in vivo support of the hypothesis that Coryln binds to Gtr1 would be meaningful. For example, it would be relatively trivial for the authors to make a *gtr1* MSN2-GFP or *gtr1* PNC1-GFP strain where the effects of corylin on Msn2 localization or Pnc1 abundance would be expected to be prevented. This experiment would also support the authors claims of Coryln mediated Tor1 dependent effects on Msn2/4.

Response:

We appreciate the reviewer's thoughtful suggestions. We agree with the reviewer that determining whether corylin increases MSN2 translocation in the *gtr1Δ* strain could provide additional support for the NMR data.

First, we designed additional peptides/controls to confirm the NMR data and verify the importance of tryptophan and isoleucine in the sequence. Please see the detailed responses to minor comments.

Second, we have included new data in Supplementary Figure.7. As suggested, we generated a *gtr1Δ*-Msn2-GFP strain and determined whether corylin increases MSN2 translocation in the *gtr1Δ* strain. We have now expanded the description in the Results section as follows: “In addition, we determined whether corylin increases MSN2 translocation in the *gtr1Δ* strain. Using fluorescence microscopy, we found that the *gtr1Δ* strain exhibited abundant MSN2 foci in the nucleus, and corylin treatment failed to further increase nuclear MSN2 accumulation in this strain (Supplementary Figure. 7).”

Additionally, if corylin has any effects on growth rate at high concentration, the *gtr1* null mutant would be expected to be resistant and the assay would be easy to perform.

Response:

As requested, we added new data after examining the effect of high corylin concentrations (30 and 60 μM) on the RLS in the WT and *gtr1Δ* strains. We found that corylin did not further increase the RLS at higher concentrations and had a similar effect as the dose used in this manuscript (15 μM).

We have modified Fig. 5G and included the following content in the results: “Furthermore, corylin was unable to increase the RLS at higher concentrations in the WT and *gtr1Δ* strains (Fig. 5g).”

The types of statistical analysis do not appear to be indicated in the paper and it is critical that these be included in methods and the figure legends.

Response:

We thank the reviewer for noting this omission. As requested, the statistical analysis methods were added to the methods section and the figure legends.

Additional comments:

The first sentences of the abstract “In long history of traditional Chinese medicine (TCM), some single herb and complex formulas have been recorded to increase lifespan in TCM pharmacopeia. However, the mechanism of these TCMS increasing lifespan is insufficient.” This reads like a factual statement and it seems likely that the lifespan extension claim is not based on scientific evidence. Stating that the mechanism is insufficient further suggests that it is. Similarly, the Table 1 title “Lists of TCM single herbs and TCM herbal formulas used to treat aging diseases and/or extend lifespan” as stated may imply that modern evidence exists for the claim.

Response:

We have changed the abstract as follows: “In the long history of traditional Chinese medicine (TCM), some single herbs and complex formulas in the TCM pharmacopeia have been suggested to increase the lifespan. However, evidence supporting the increase in the lifespan is lacking in the TCM pharmacopeia.”

The title of table 1 has been changed to “Lists of TCM single herbs and herbal formulas in the TCM pharmacopeia that are suggested to treat age-related diseases and/or extend the lifespan”

Line 64-65 “Despite such extensive studies in the field of aging, the number of compounds that extend lifespan is very small”. There are numerous compounds that have been published to extend the lifespan of various model organisms, perhaps this sentence should be worded something like “the number of compounds identified by the interventions testing program that extend the lifespan of mice is very small” or something like rigorously identified to extend lifespan in mice. A similar statement is made at the beginning of the results section (line 124: “only a few compounds have been shown to increase lifespan”).

Response:

As suggested, we have rewritten the sentence as follows:

1. “Despite such extensive studies in the field of aging, the number of compounds identified by intervention testing programs that extend the lifespan in mice is very small.”

2. “To date, only a few compounds rigorously identified to extend lifespan.”

Lines 70-72, “Of the available screening methods for lifespan-extending drugs, the SIRT1 in vitro assay is the most common method, as it allows relatively simple and fast data collection”. I don’t think this is the most common method of screening for lifespan-extending drugs, perhaps more appropriate would be “one method” instead of “most common method”.

Response:

We appreciate the reviewer’s thoughtful suggestions, and we have rewritten the sentence as follows: “Of the available screening methods for lifespan-extending drugs, the SIRT1 in vitro assay allows relatively simple and rapid data collection.”

In the results section, it should be indicated that the other 77 candidates presumably did not show evidence for lifespan extension under the conditions tested. The concentrations tested and extraction method for the other extracts could perhaps be noted in the event other researchers found the information useful.

Response:

We appreciate the reviewer’s thoughtful suggestion. We have added new data to supplementary Figure. 2 and Supplementary Figure. 3.

Lines 212-214 “Additionally, several studies have shown that inhibiting tor1 relocates Msn2/4 from the cytoplasm to the nucleus, and promotes PNC1 expression [22].” The sentence states “Several studies”, yet only one is referenced.

Response:

As suggested, the sentence has been corrected as follows: “Additionally, a previous study showed that inhibiting tor1 relocates Msn2/4 from the cytoplasm to the nucleus and promotes PNC1 expression.”

“Pnc1 hydrolyzes nicotinamide to nicotinic acid as a precursor in NAD⁺ salvage, which increases NAD⁺ levels, activating stress responses and increasing lifespan”. Has Pnc1 been shown to increase lifespan? The reference is a review that references a paper simply showing the deletion of PNC1 prevents lifespan extension by dietary restriction (glucose reduction). Msn2/4 localizes to the nucleus in response to many different stressors and the relation to Tor1 in Figure 4 is inferred but is not directly

supported by any evidence. Increased Msn2-GFP nuclear localization being due to corylin regulation of Tor1 seems like an overly strong conclusion based on the data in this figure. Given Pnc1 is a Msn2/4 target, this only supports the observed effect with Msn2/4. The experiments proposed above would yield more direct support.

Response:

1. We thank the reviewer for noting this omission. Pnc1 has been shown to increase the lifespan of yeast, and the reference has been corrected in the revised manuscript.
2. As discussed above, we generated a *gtr1Δ*-Msn2-GFP strain, and we have included new data in Supplementary Figure. 7.

Lines 226-28 “As shown in Figure 4H, NAD⁺ levels were significantly increased following corylin treatment, suggesting that *tor1* signaling was inhibited by corylin treatment.” Does *tor1* deletion increase NAD⁺ levels? This correlation would seem to be necessary for the argument and would likely be known and could be referenced.

Response:

Indeed, a previous report implies a relationship between *tor1Δ* and NAD⁺. However, there is no direct evidence of this link. As requested, we have rewritten the sentence as follows: “As shown in Fig. 4h, NAD⁺ levels were significantly increased following corylin treatment., ~~suggesting that *tor1* signaling was inhibited by corylin treatment.~~”

How was nuclear localization of Msn2-GFP scored? E.g., manually or some threshold for puncta? Should be included in methods.

Response:

As requested, the following description has been added to the methods section: “Nuclei were stained with Hoechst #33342 to identify live cells. The number of cells with nuclear Msn2-GFP was counted manually and normalized to those with Hoechst staining.”

What was the vehicle for corylin and what was the control condition used (equal concentration of vehicle only?). Generally it is advisable to be sure the vehicle has no effect by including an additional control with untreated (no vehicle).

Response:

1. The vehicle in this study was DMSO, and the concentration was 2 μl/ml for the yeast experiment and 1 μl/ml for cell experiments.
2. As suggested, an untreated HUVEC group has been incorporated into Fig. 6a.

Figure 1. The legend indicates a “10 µg/ml ethanol extract of *P.corylifolia* or DMSO”, but the figure itself indicates ethanol extract of “Fructus Psoraleae”. This seems an odd error.

n-HEAXEN instead of n-hexane is included in panels 1D,E, and F.

Response:

We thank the reviewer for clarifying these errors, and we have changed the terms to “*P. corylifolia*” and “n-HEAXEN” in the revised manuscript.

Figure 6a, was a no DMSO control also performed?

Response:

As discussed above, we have incorporated an untreated group (no DMSO) in Fig. 6a.

Figure 5b and 5c, The 0 and 5 indications are slightly ambiguous. It appears to indicate the µM concentration of corylin, but that should be more clearly indicated in the legend for the panel. It may be easier for a reader to interpret G and H if “S/Y” and “S+C/S” were defined in the legend.

Response:

As suggested, the labels for Fig. 5b and 5c were added to the figure, and the descriptions of “S/Y” and “S+C/S” were added to the figure legend to clarify the ambiguity.

Interestingly, the mice food intake was unchanged. Rapamycin (an mTOR inhibitor) reduced food intake in HFD C57BL/6 mice (Geng-Tuei et al. J. Pharm. Sci 2009). This would suggest differences in when intervening against Gtr1 vs mTorc1 (assuming similar HFD was performed). Is it known that rapamycin or mtorc1 knockdown has similar effects to corylin on glucose and cholesterol in mice fed HFD? Comparisons of mtor1 knockouts and rapamycin with HFD from literature here would be useful.

Response:

It is not clear that mTOR1 knockdown has longevity effects similar to those of corylin treatment on glucose and cholesterol in mice fed an HFD. We have compared corylin and rapamycin and expanded the description in the Discussion. “Rapamycin, an mTOR1 inhibitor, has been shown to increase the lifespan and decrease weight gain in

a HFD model. However, rapamycin elicited hyperglycemic effects via mTOR2 inhibition with long-term treatment, which could be reversed by rapamycin discontinuation(Harrison et al., 2009; Jung et al., 2016). This indicates that universal inhibition of mTOR signaling could generate side effects (Blagosklonny, 2019). Although the body weight was similar to that of the HFD group following corylin treatment, the fasting blood glucose level was lower than that of the HFD group. Furthermore, we found that corylin decreased TGs in HFD-fed mice, which was not observed with rapamycin(Kenerson, Yeh, & Yeung, 2011). Taken together, these results suggested that inhibiting/partly inhibiting mTOR1 signaling by Rag A but not mTOR2 could be an attractive intervention for increasing lifespan.”

For the NMR data, were coupling constants determined? Wouldn't the corylin aromatic peaks in the spectrum also be expected to shift? Rather than remove the two residues thought to be important for the peptide binding to corylin and add two residues at the N and C terminal ends, the substitution of the residues would have seemed like a better control peptide. And perhaps a peptide with tryptophan and isoleucine at different locations may be meaningful in terms of the specificity of the interaction. It is being assumed that the small peptide is taking on a structure similar to that observed within the folded protein, but the simplicity of using a peptide to model a part of the folded protein makes interpretation of the results difficult.

Response:

1. None of the corylin peaks in the proton spectrum shifted; hence, the coupling constants were not calculated. Moreover, ¹H NMR spectra were added to Supplementary Fig. 6.
2. As requested, new data were added. We designed additional peptides to strengthen our hypothesis:
 - (1) Tryptophan and isoleucine were substituted with glutamine
 - (2) Tryptophan and isoleucine were relocated.

The NMR results showed that the substitution of residues (1) resulted in no shifts in the presence of corylin, and the relocated peptide (2) showed minor shifts in the spectrum with corylin treatment. This result suggested that tryptophan and isoleucine are crucial for corylin interaction and that the location of tryptophan and isoleucine might be important for the interaction of corylin.

We have expanded the description in the Results section: ”To confirm the direct interactions between GTR1 and corylin, ¹H NMR chemical shift experiments were

conducted (Supplementary Fig. 6). Four different peptides were designed as probes to investigate the interaction between GTR1 and corylin. (i) peptide 1: WT sequence, (ii) peptide 2: ile166 and trp167 were substituted with glutamine, (iii) peptide 3: ile166 and trp167 were omitted from the sequence, and (iv) peptide 4: ile166 and trp167 were relocated in the sequence (Fig. 5c). Meanwhile, we found that GTR1 (Rag A in mammals) is highly conserved across species, especially the corylin binding region that we proposed (Fig. 5d). As shown in Fig. 5e, peptide 1 showed upfield resonance at 8.03, 8.06, 7.87, and 7.84 ppm and downfield resonance at 7.92, 7.94, and 8.13 ppm upon additional corylin treatment. Interestingly, peptides 2 and 3, which lacked ile166 and trp167, showed a zero shift in the presence of corylin. In addition, peptide 4, in which ile166 and trp167 were relocated, showed upfield resonance at 8.1 and 7.7 ppm. This result suggested that the ile166 and trp167 residues of GTR1 are essential for corylin binding, while the location of tryptophan and isoleucine might be important for the interaction of corylin.”

The discussion ends “with strong evidence showing the lifespan extension properties of corylin in multiple organisms”. One could argue that normal lifespan extension was only shown in yeast, and the HFD data in mice shows that corylin provides some stress resistance in this organism.

Response:

We appreciate the reviewer’s thoughtful suggestion, and we have now changed the sentence to “with strong evidence showing the lifespan extension properties of corylin in yeast and cell models and improving health and survival by facilitating stress resistance in HFD mice.”

In conclusion, we are extremely grateful for the reviewer’s time, care, and constructive suggestions. The manuscript has been refined follow by the suggestion above, which we feel the mechanistic insight from this study has been drastically improved.

REVIEWER COMMENTS

Reviewer #1 (Remarks to the Author):

The authors have now addressed all my comments concerns about the manuscript

Reviewer #2 (Remarks to the Author):

Overall, the authors appear to have reasonably addressed the reviewer comments. I have only very minor additional comments.

The manuscript title is vague and a more specific title might be warranted.

The last sentence of the abstract could use an additional qualifier, "Taken together, these findings demonstrate that corylin has long-term benefits for longevity and could be a potential treatment for first age-related diseases." It could perhaps read 'corylin may have benefits for longevity' or 'benefits for longevity in yeast'. It's also not clear to me what a "first" age-related disease is.

Reviewer #3 (Remarks to the Author):

Validation of bioactive components from traditional Chinese medicine for lifespan extension
NCOMMS-20-48816A

This report is related to the section: Corylin increases yeast RLS by targeting the GTR1 protein

In this section, the authors describe the identification of GTR1 protein as the potential target for Corylin.

From the text, the identification of the protein as a potential target for this compound is unclear. Could the authors explain how this target -among the many possible proteins participating in the TOR1 signalling pathway- was identified?

Was this identification made using molecular docking as it seems to be described?

Which controls were made to ensure that this protein is indeed the genuine target and the one and only based on ligand docking? The details explaining how the target selection and docking were performed are not explained in the results section.

Then, assuming that the target selection is fully supported by data, there is no experimental control with the full length GTR1 protein, to validate that a compound-ligand interaction occurs in vitro. In the absence of this control, the assumption that this potential interaction is the same as that of the compound with the short peptides that seem to map the binding region seems very risky.

Assuming that the fragments behave as the full-length protein, the binding experiments were carried out in DMSO, which does not seem the most suitable solvent for comparing the binding results of the compound and the protein in native conditions. Did the authors compare that the peptides in H₂O and DMSO behave the same before setting up the binding experiments?

Moreover, it seems like the authors did not assign the NMR data corresponding to the peptides (the assignment is not provided). In the absence of this information, the critical role of the Trp and Ile residues involved in the hypothetical binding is not supported.

The authors also claimed that both Trp and Ile residues are essential for the hypothetical interaction. However, in the 1D superposition, no changes are observed in the methyl region, to support the participation of the Ile residue in this interaction.

The choice of 1D-NMR to identify possible interactions is risky given the level of Chemical shift overlap at 400 MHz. Perhaps the authors should consider to acquire 2D 1H-NMR to overcome this

limitation. Independently of the use of 1D or 2D NMR, the lack of proper controls with another flavonoid with a Corylin-related skeleton made difficult to argue that the Chemical shift perturbations indicated are not due to non-specific interactions between aromatic rings in DMSO. As previously reported by one of the reviewers, it is highly surprising that the Corylin compound, despite interacting, does not show any chemical shift perturbation upon "binding".

Overall, the entire section describing the target identification and the interaction of the compound with the protein as it is now, is weak and it will need a profound revision with proper controls using the FL protein and other Corylin related compounds.

If this section is not essential to support the main conclusions of the manuscript, perhaps the authors should consider removing it.

Black: our response

Underlined: changed in our manuscript. In the revised manuscript, we also highlighted changes in green. The previously revised text is highlighted in red.

Dear Editor and Reviewers,

We are submitting a revised version of our manuscript (NCOMMS-20-48816-A), previously entitled "Validation of bioactive components from traditional Chinese medicine for lifespan extension".

We thank the reviewers for their extremely careful and constructive reviews of our work. We have provided point-by-point responses to the reviewers' specific comments below.

REVIEWER COMMENTS

Reviewer #1 (Remarks to the Author):

The authors have now addressed all my comments concerns about the manuscript

Response:

Thank you for the Reviewer's comments and time, we fell manuscript has been large improved after the revision.

Reviewer #2 (Remarks to the Author):

Overall, the authors appear to have reasonably addressed the reviewer comments. I have only very minor additional comments.

Response:

We thank the Reviewer for the helpful suggestions that making our manuscript a more rigorous of work.

The manuscript title is vague and a more specific title might be warranted.

Response:

We appreciate the reviewer's thoughtful suggestions. The title has now been changed to Validation of lifespan extension components from traditional Chinese medicine: corylin, an active compound for longevity.

The last sentence of the abstract could use an additional qualifier, "Taken together, these findings demonstrate that corylin has long-term benefits for longevity and could be a potential treatment for first age-related diseases." It could perhaps read 'corylin may have benefits for longevity' or 'benefits for longevity in yeast'. It's also not clear to me what a "first" age-related disease is.

Response:

We apologize for the mistake: the word "first" has been removed, and the last sentence of the abstract has been changed to Taken together, these findings demonstrate that corylin may promote longevity.

Reviewer #3 (Remarks to the Author):

This report is related to the section: Corylin increases yeast RLS by targeting the GTR1 protein

In this section, the authors describe the identification of GTR1 protein as the potential target for Corylin.

From the text, the identification of the protein as a potential target for this compound is unclear. Could the authors explain how this target -among the many possible proteins participating in the TOR1 signalling pathway- was identified?

Was this identification made using molecular docking as it seems to be described?

Which controls were made to ensure that this protein is indeed the genuine target and the one and only based on ligand docking? The details explaining how the target selection and docking were performed are not explained in the results section.

Response:

Several decisions were made in the process of identifying potential targets of corylin before we conducted molecular docking. First, we used RLS to verify two major pathways, the Sir2 and Tor1-dependent pathways, that are crucial for yeast RLS. We found that corylin is related to the Tor1-dependent pathway; more specifically, Tor1 is required for RLS extension upon corylin treatment in yeast. Next, we also noticed that the RNA levels of the mTor pathway were significantly changed following corylin treatment in HUVECs. This suggests that corylin targets the upstream proteins of Tor1.

As requested, some descriptions have been added to the results. Since corylin increases RLS in a Tor1-dependent manner, we tested possible targets involved in the Tor pathway in yeast by using molecular docking software, and the docking results were displayed by Discover Studio based the on the references.

Then, assuming that the target selection is fully supported by data, there is no experimental control with the full length GTR1 protein, to validate that a compound-ligand interaction occurs in vitro. In the absence of this control, the assumption that this potential interaction is the same as that of the compound with the short peptides that seem to map the binding region seems very risky.

Response:

The conclusion that corylin interacts with short peptides instead of full length Gtr1

protein in vitro would indeed seem risky without supporting data. However, in the first revision of this manuscript, we added new data and showed that the deletion of Gtr1 increases RLS and that corylin no longer increases RLS in the gtr1 mutant (figure 5I). Moreover, Gtr1 overexpression decreased RLS in yeast, and corylin failed to rescue RLS in the strain (figure 5J), which shows that corylin genetically increased yeast RLS by targeting the Gtr1 protein.

Most importantly, we noticed that these short peptides are highly conserved among species. This suggests that the peptide region is important and could provide additional information among different species for using these conserved peptides but not full-length peptides in NMR experiments. After all, it is still possible that the interaction data of corylin with the full-length Gtr1 protein of yeast may still be unable to represent other species.

We agree with the reviewer that the binding experiment of corylin with full-length protein in vitro is important, and we are planning to perform this experiment using full-length Rag A (Gtr1 in yeast) in the future.

Assuming that the fragments behave as the full-length protein, the binding experiments were carried out in DMSO, which does not seem the most suitable solvent for comparing the binding results of the compound and the protein in native conditions. Did the authors compare that the peptides in H₂O and DMSO behave the same before setting up the binding experiments?

Response:

We appreciate the reviewer's thoughtful suggestions. We have previously tried to dissolve corylin in H₂O and found that corylin is insoluble in H₂O, hence it is less likely to use H₂O as a solvent in the experiment.

Moreover, it seems like the authors did not assign the NMR data corresponding to the peptides (the assignment is not provided). In the absence of this information, the critical role of the Trp and Ile residues involved in the hypothetical binding is not supported.

Response:

As suggested, we tried to assign peptide 1 by using 1D and 2D 600 MHz NMR. However, due to the similarity of some amino acids, many signals overlap, and it is technically difficult to assign the peptide. For reference, we still added the ¹H, ¹³C, DEPT135, COSY1, HSQC, and HMBC spectra of peptide 1 (Supplementary Fig. 14). Moreover, the synthesis report of peptide 1 from the outsourcing company is included in supplementary data figure 15.

Although we did not assign the peptides, we provide several controls, as other

reviewers suggested, to strengthen our hypothesis

1. Peptide 2: Ile166 and Trp167 replaced with Gln showed no showed no signal shifts in the presence of corylin.
2. Peptide 3: Ile166 and Trp167 removed showed no showed no signal shifts in the presence of corylin.
3. Peptide 4: Ile166 and Trp167 were relocated, showed weak signal shifts in the presence of corylin.

This result suggested that the Ile166 and Trp167 residues of GTR1 are essential for corylin interaction, while the locations of tryptophan and isoleucine might be important for the interaction of corylin.

The authors also claimed that both Trp and Ile residues are essential for the hypothetical interaction. However, in the 1D superposition, no changes are observed in the methyl region, to support the participation of the Ile residue in this interaction.

Response:

Based on the docking result, the bonds are all Pi interactions and hydrogen bonds. The NMR spectra showed no actual binding, i.e., hydrogen bonds, which suggests that the cause of the peptide perturbation was Pi interaction. Hence, the aromatic rings of corylin might interact with amine groups in the peptide, causing the methyl groups to be distant from the aromatic rings of corylin. Perhaps that is why no change in the methyl groups was observed.

The choice of 1D-NMR to identify possible interactions is risky given the level of Chemical shift overlap at 400 MHz. Perhaps the authors should consider to acquire 2D 1H-NMR to overcome this limitation. Independently of the use of 1D or 2D NMR, the lack of proper controls with another flavonoid with a Corylin-related skeleton made difficult to argue that the Chemical shift perturbations indicated are not due to non-specific interactions between aromatic rings in DMSO.

Response:

We appreciate the reviewer's thoughtful suggestion. We added new data to address the nonspecific/specific binding issue. We selected daidzein as a control, which shares the main structure with corylin, and we showed that daidzein did not exhibit lifespan extension properties in our MEP system (figure 3E).

We found that peptide 1 showed weak signal shifts upon daidzen treatment and strong signal shifts upon corylin treatment in the ¹H NMR spectrum (Supplementary Fig. 7). This result indicates that the aromatic rings on both corylin and daidzein could cause chemical shift perturbations, but it also supports our hypothesis that corylin specifically interacts with peptide 1 since the signal shifts are much stronger in the

presence of corylin than in the presence of daidzen (Fig. 5e and Supplementary Fig. 7).

We have added new data and included the following content in the results: As shown in Fig. 5e, peptide 1 showed strong signal shifts at 8.13, 8.0, 7.94, 7.9, 7.86 and 7.8 ppm upon corylin treatment. Interestingly, peptide 2 and 3, which lacked Ile166 and Trp167, showed no signal shifts in the presence of corylin. In addition, peptide 4, in which Ile166 and Trp167 were relocated, showed weak signal shifts at 8.1 and 7.7 ppm. This result suggested that the Ile166 and Trp167 residues of Gtr1 are essential for corylin interaction, while the locations of Ile166 and Trp167 might be important for the interaction of corylin. Moreover, we performed a control with daidzen, which shares the major skeleton with corylin and did not increase the RLS in the MEP assay. We found that peptide 1 showed weak signal shifts upon daidzen treatment and strong signal shifts upon corylin treatment in the ¹H NMR spectrum (Supplementary Fig. 7). This result indicates that the aromatic rings on both corylin and daidzein could cause chemical shift perturbations, but it also supports our hypothesis that corylin specifically interacts with peptide 1 since the signal shifts are much stronger in the presence of corylin than in the presence of daidzen (Fig. 5e and Supplementary Fig. 7).

As previously reported by one of the reviewers, it is highly surprising that the Corylin compound, despite interacting, does not show any chemical shift perturbation upon "binding".

Response:

Assuming that the unchanged methyl groups are in line with our hypothesis (described above), the aromatic rings of corylin interact with amino groups in the peptide. Hence, the electronic clouds on the aromatic rings of corylin could draw amino groups to form Pi interactions, but the amino groups did not generate interactions that affected corylin.

Overall, the entire section describing the target identification and the interaction of the compound with the protein as it is now, is weak and it will need a profound revision with proper controls using the FL protein and other Corylin related compounds.

If this section is not essential to support the main conclusions of the manuscript, perhaps the authors should consider removing it.

Response:

In summary, the NMR experiment on the interactions of corylin with the conserved peptides combined with the supporting data (described above) could still provide some information among different species.

In conclusion, we are extremely grateful to all the reviewers for their time, care and constructive suggestions. We feel that the significance and mechanistic insight from this study has been drastically improved by their advice.

REVIEWER COMMENTS

Reviewer #2 (Remarks to the Author):

The authors appear to have reasonably addressed my review comments.

Reviewer #3 (Remarks to the Author):

I appreciate very much the efforts made by the authors to reply to the queries.

Still, I believe that the section describing corylin binding to peptides is dubious. The interaction is not supported by any kind of affinity value or even docking scores. In the absence of a positive control with the GTr1 protein, I am not sure that this section is solid and permits to assume that Corylin binds specifically to the Gtr1 protein.

The authors should rephrase the following part and explain why they have discarded the other two binding sites. Could the authors display this information together with the binding scores? Also a superposition of the "30 poses"?

Line 246 "The crystal structure of yeast Gtr1-Gtr2 protein (PDB id: 3R7W) was docked with corylin (Fig. 5a). The docking analysis showed 3 different binding domains with the Gtr1 protein, two of which were with the N-terminus, which is the GTPase-active domain. Furthermore, over 30 poses were generated, and based on bonding type, bonding distance and the position of the corylin structure, we selected possible results.

Black: our response

Underlined: changed in our manuscript. In the revised manuscript, we highlighted changes in blue. The previously revised text is highlighted in red (NCOMMS-20-48816A) or green (NCOMMS-20-48816B).

Dear Editor and Reviewers,

We are submitting a revised version of our manuscript entitled “Validation of lifespan extension components from traditional Chinese medicine: corylin, an active compound for longevity”.

We thank the reviewers for their extremely careful and constructive reviews of our work. We have provided point-by-point responses to the reviewers’ specific comments below.

REVIEWER COMMENTS

Reviewer #2 (Remarks to the Author):

The authors appear to have reasonably addressed my review comments.

Response:

Thank you for your efforts to help us improve the quality of our manuscript.

Reviewer #3 (Remarks to the Author):

I appreciate very much the efforts made by the authors to reply to the queries. Still, I believe that the section describing corylin binding to peptides is dubious. The interaction is not supported by any kind of affinity value or even docking scores. In the absence of a positive control with the GTr1 protein, I am not sure that this section is solid and permits to assume that Corylin binds specifically to the Gtr1 protein.

Response:

We appreciate the reviewer’s thoughtful comments.

In this section, we aim to emphasize that “corylin increases yeast RLS by targeting the Gtr1 protein” instead of “corylin bind to Gtr1” *per se*. After incorporating new experiments (listed below) that suggested by reviewers, we feel that the statement has drastically improved.

1. Corylin genetically increased yeast RLS by targeting the Gtr1 protein (suggested by reviewer#2). In the previously revised manuscript, we showed that the deletion of GTR1 increases RLS and that corylin no longer increases RLS in the *gtr1* mutant (figure 5I). Moreover, Gtr1 overexpression decreased RLS in yeast, and corylin failed to rescue RLS in the strain (figure 5J)
2. Corylin fail to further increase nuclear MSN2 accumulation in the *gtr1Δ* strain

(suggested by reviewer#2). We found out that the *gtr1Δ* strain exhibited abundant MSN2 foci in the nucleus, and corylin fail to further increase nuclear MSN2 accumulation in this strain (Supplementary Fig. 9).

3. The interaction region of GTR1 peptides is highly conserved among species.
4. The Ile166 and Trp167 residues and the locations of Ile166 and Trp167 of Gtr1 might be important for the interaction of corylin (suggested by reviewer#2). Four different peptides were designed as probes to investigate the interaction between Gtr1 and corylin and the data were shown in Figure 5C
5. The conserved peptides interact more specifically with corylin than with daidzein (suggested by reviewer#3). We thank the reviewer for asking the control, and the data demonstrate that corylin does not just randomly interact with the Gtr1 peptides (Supplementary Fig. 8).

We also understand that the phrase “specific binding” or “binding” may over-interpret the data. Therefore, we also rephrase some of the sentence in this section to make them clear and less confusing for readers.

1. To confirm the direct interactions between Gtr1 and corylin.....
2. “This result suggested that the Ile166 and Trp167 residues of Gtr1 are essential for corylin interaction, while 264 the locations of Ile166 and Trp167 might be important for the interaction of corylin.” has now been changed to “This result suggested that the residues and locations of Ile166 and Trp167 might be important for the interaction of corylin.
3.,”but it also supports our hypothesis that corylin specifically interacts with peptide 1 since”... has now been changed to “and it also indicates that peptide 1 interacts more specifically with corylin than daidzein since”...

The authors should rephrase the following part and explain why they have discarded the other two binding sites. Could the authors display this information together with the binding scores? Also a superposition of the "30 poses"?

Line 246 "The crystal structure of yeast Gtr1-Gtr2 protein (PDB id: 3R7W) was docked with corylin (Fig. 5a). The docking analysis showed 3 different binding domains with the Gtr1 protein, two of which were with the N-terminus, which is the GTPase-active domain. Furthermore, over 30 poses were generated, and based on bonding type, bonding distance and the position of the corylin structure, we selected possible results.

Response:

As requested, we have provided the docking analysis of 3 different interaction domains with the Gtr1 in Supplementary Fig 6. We also rewrote this paragraph to make the statements straightforward and less confusing. “Accordingly, we found that

Gtr1 is a potential target of corylin. The crystal structure of yeast Gtr1-Gtr2 protein (PDB id: 3R7W) was docked with corylin (Fig. 5a). The docking analysis showed 3 different binding domains with the Gtr1 protein, two of which were with the N-terminus, which is the GTPase-active domain. Furthermore, over 30 poses were generated, and based on bonding type, bonding distance and the position of the corylin structure, we selected possible results. Among the results, in many poses, Ile166 and Trp167 showed potential binding with corylin (Fig. 5b)” has now been changed to “By the docking analysis, we found that Gtr1 is a potential target of corylin (Fig. 5a). Furthermore, the docking analysis demonstrated that three different domains may interact with corylin, and one of the interaction domains which contain Ile166 and Trp167 showed stronger interaction with corylin (Fig. 5b and supplementary Fig 6).”

Supplementary Figure 6 The comparison of 3 different domains that may interact with corylin by the docking analysis. (A-C) The CDOCKER-interaction-energy and superpositions of corylin with Gtr1 (PDB ID: 3R7W). The interaction domains of (A) which contain Ile166 and Trp167 showed stronger interaction with corylin.

In conclusion, we are extremely grateful for the reviewer’s time, care, and constructive suggestions. We have now incorporated all of the above-mentioned changes in the text to refine our manuscript, which we feel the manuscript has

drastically improved by the advice.

REVIEWERS' COMMENTS

Reviewer #3 (Remarks to the Author):

The authors appear to have reasonably addressed my review comments.